# Flame-retardant electrolytes with electrochemically-inert and weakly coordinating dichloroalkane diluents for practical lithium metal batteries

Zhicheng Wang[1,2,8], Haifeng Tu[3,8], Xingdong Ma[1], Suwan Lu[3], Guirong Su[4], Yiwen Gao[3], Jiangyan Xue[3], Lingwang Liu[3], Xu Yao[1], Kun Liang[1], Ke Wang[5], Fengrui Zhang[1,2], Zhifeng Qin[1], Jieyun Zheng[1,2], Qing Wang[6], Jingjing Xu[3,4,5,9] ✉, Liquan Chen[1,2], Hong Li[1,2,7,9] ✉ & Xiaodong Wu[1,3,5,9] ✉

The next-generation of lithium metal batteries urgently require electrolytes that simultaneously possess low-cost, high-safety, wide-temperature operating range, high electrochemical stability and good electrode-electrolyte interphases formation ability. Here we present a flame-retardant electrolyte by introducing electrochemically-inert and weakly coordinating dichloroalkane diluents in triethyl phosphate-based high-concentration electrolyte. We systematically investigate the effects of dichloroalkane diluents with diverse carbon chain lengths on the $Li^+$ solvation structure, redox behavior, and lithium metal interfacial chemistry in the electrolyte. Consequently, 1,3-dichloropropane, which shows the favorable electrochemical inertness, weakly coordinating ability and wide liquid temperature range (−99 to +120 °C), is chosen as an ideal diluent in electrolyte to form robust anions-derived inorganic-rich electrode-electrolyte interphases on electrodes and improve the $Li^+$ transport/de-solvation capability. The developed electrolyte exhibits significant improvement in safety, cycling stability, rate capability and wide temperature operation capability of high-voltage lithium metal batteries. Particularly, the practical Li (50 µm)||$LiNi_{0.83}Co_{0.12}Mn_{0.05}O_2$ (NCM83, 5.6 mAh cm$^{-2}$) pouch cells exhibit stable cycling performance over 100 cycles with a high capacity retention rate of 94.1% at 0.1 C charge/0.2 C discharge under 25 °C, and deliver a promising application potential within a broad temperature range of −60 to +60 °C.

Lithium (Li) metal batteries (LMBs), assembled with Li metal negative electrode and high-voltage nickel (Ni)-rich positive electrode, are recognized as one of the most promising next-generation high-energy-density energy storage devices[1–3]. Nonetheless, their practical deployment is impeded by challenges such as the limited cycle life attributed to the high reactivity of Li metal and safety concerns arising from Li dendrite formation[4,5]. The electrolyte, an essential component of the battery, significantly influences the battery's electrochemical performance[6,7]. Regrettably, the current state-of-the-art commercial electrolytes based on carbonate esters, characterized by their narrow

electrochemical stability window (~4.3 V), restricted operational temperature range (−20 to +50 °C), and high flammability, pose significant constraints on the electrochemical performance and safety of LMBs[8–10].

Recent research has demonstrated that innovative electrolyte formulations can markedly enhance the cycle stability and safety of LMBs[11–14]. For example, phosphate ester solvents endowed with commendable flame-retardant properties, such as trimethyl phosphate (TMP) and triethyl phosphate (TEP), have been extensively employed as principal solvents in electrolytes[15–18]. By modulating the electrolyte solvation structure through the incorporation of high-concentration lithium bis(fluorosulfonyl)imide (LiFSI) salts, the proportion of free solvent within the system is diminished, effectively curtailing interfacial side reactions between phosphate ester solvents and Li metal electrode[16]. This approach has been successfully implemented in LMBs to yield enhanced cycle stability and safety[17,18]. Nevertheless, these high-concentration electrolytes (HCEs) also suffer from many issues, such as high viscosity, low ionic conductivity, and high cost, which severely restrict the application of LMBs, especially under low-temperature and high-rate conditions[19–21]. To ameliorate the physicochemical properties of HCEs, weakly polar fluorinated ether solvents with low viscosity and electrochemical inertness, often termed non-coordinating diluents[22,23], have been integrated into HCEs to create local high-concentration electrolytes (LHCEs), such as bis(2,2,2-trifluoroethyl) ether (BTFE)[24,25], 1,1,2,2-tetrafluoroethyl-2,2,3,3-tetrafluoropropyl ether (TTE)[26,27], 3,3,4,4,5,5-hexafluorotetrahydropyran (HFTHP)[28]. These highly fluorinated diluents, incapable of dissolving Li salts, exhibit minimal interaction with Li+, thereby preserving an anions-dominated Li+ solvation structure and fostering the formation of anions-derived inorganic-rich electrode-electrolyte interphases (EEIs) on both negative electrode and positive electrode surfaces[27,28]. These LHCEs effectively suppress the Li dendrite growth and enhance the cycle life and rate capability of LMBs. However, the limitations of these highly fluorinated inert diluents, including their restricted Li+ transport kinetics, elevated cost, and increased density, especially be prohibited by many countries due to their potential harm to human and environment, render them less suitable for practical application[29,30]. Consequently, there is a pressing requirement to develop novel diluents to optimize the design of LHCEs and facilitate their application in LMBs with superior performance over a wide temperature range.

The design idea of electrolyte diluent in this work is depicted in Fig. 1a. Firstly, the primary criteria for diluent selection should be suitable physicochemical properties of diluent, such as low freezing point, high boiling point, low viscosity, non-flammability, and low cost. Secondly, the weakly coordinating ability and high electrochemical inertness set the secondary criteria for diluent selection, which is expected to enhance the Li+ transport/de-solvation kinetics by partially coordinated with Li+, while easily inducing the preferential decomposition of anions due to the electrochemical inertness, thereby

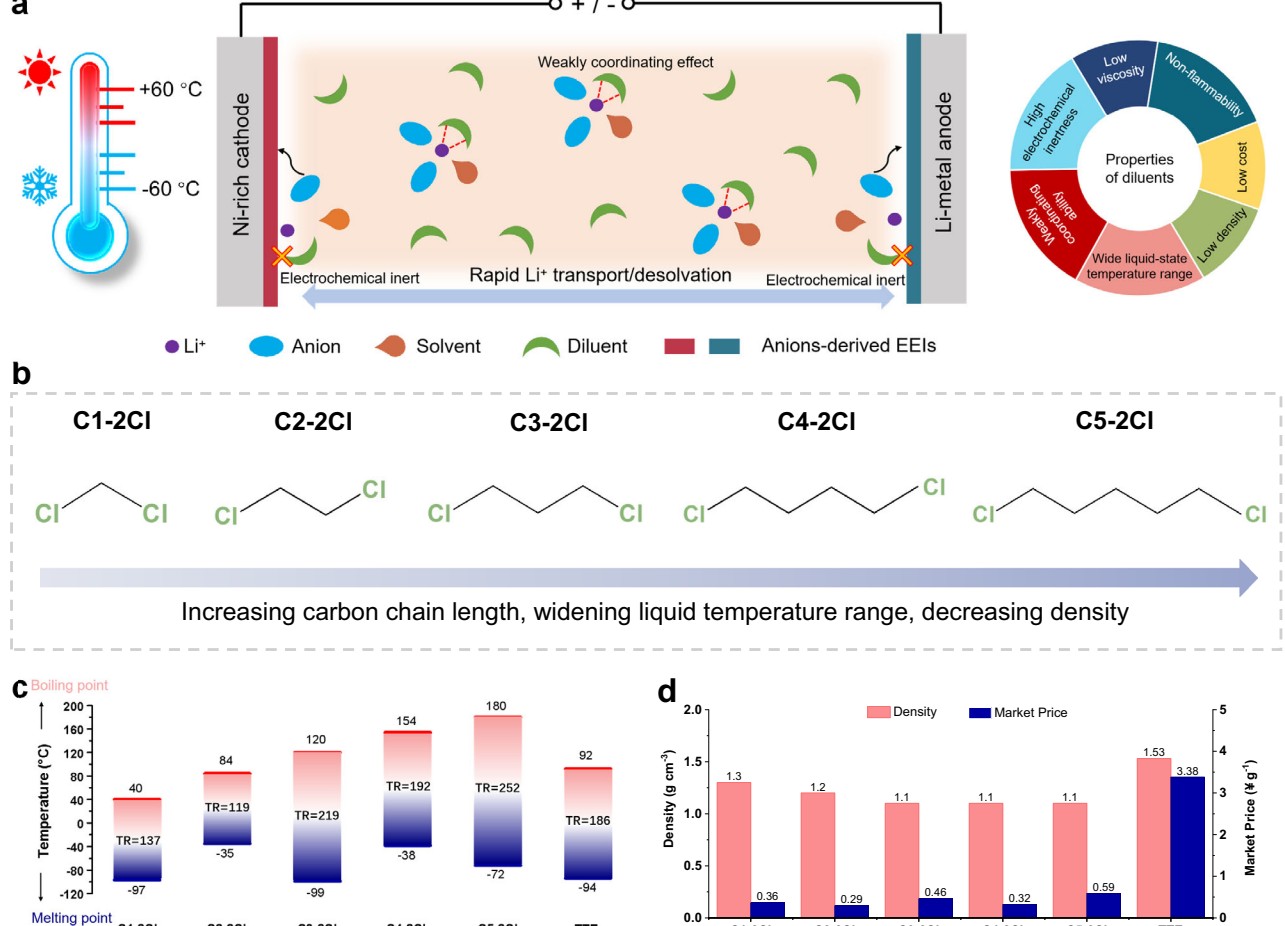

**Fig. 1 | Design of ideal electrolyte diluents for wide-temperature LMBs. a** Illustration of the electrolyte solvation structure, rapid Li+ transport/de-solvation, electrodes interfacial chemistry, and properties of diluents in the flame-retardant electrolyte with C-2Cl diluents. **b** Molecular structures of C-2Cl diluents with increased carbon chain lengths. **c** Boiling and melting points of different diluents. **d** Density and market price of different diluents. The market price is calculated based on the price of 1 kg of the Adamas brand product from ref. 58 on Sep. 1st, 2024.

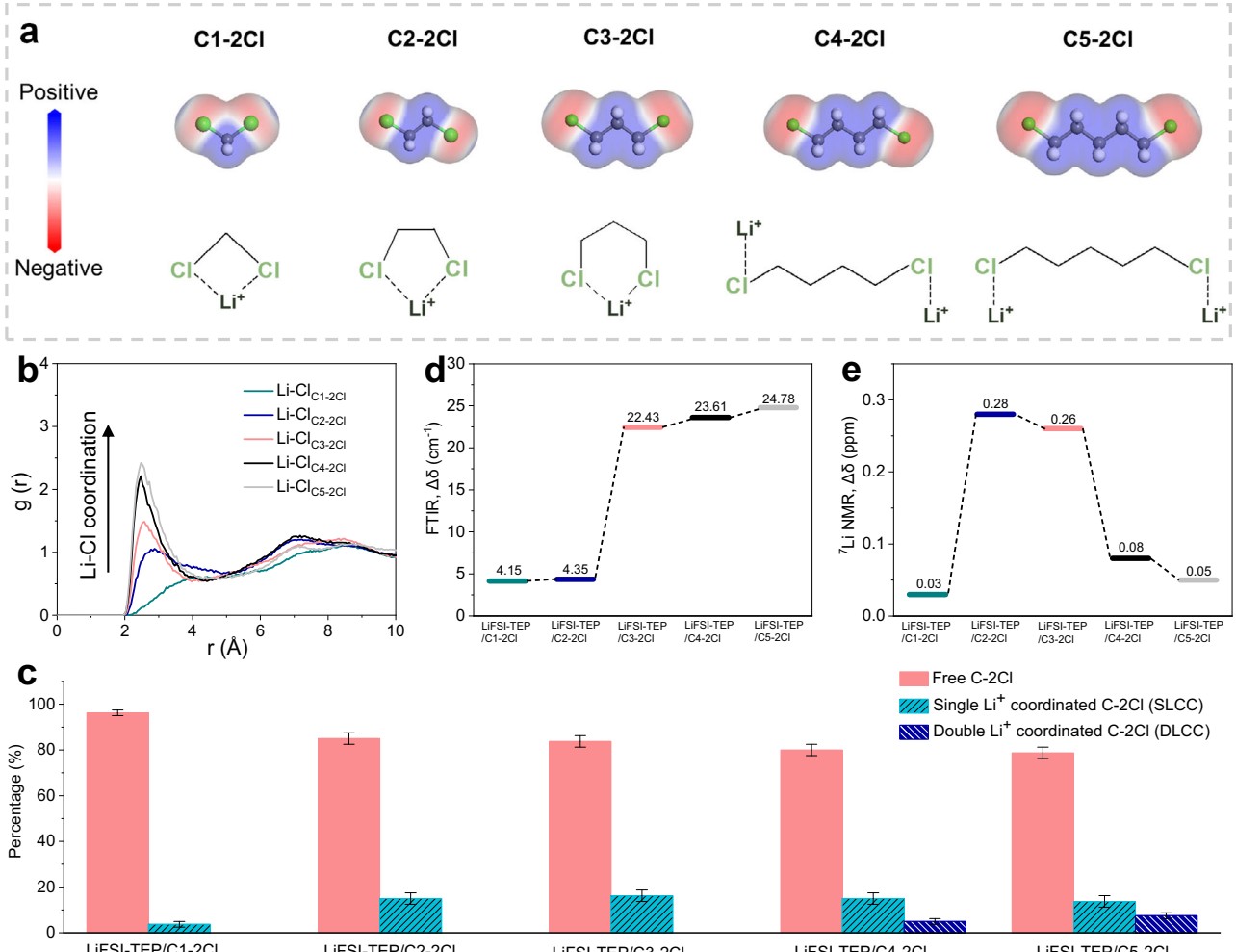

**Fig. 2 | Analysis of the electrolyte solvation structures with different C-2Cl diluents. a** ESP of different C-2Cl diluents and the coordination structures of Li⁺ with different C-2Cl diluents. Color code: C-dark gray, H-white, Cl-green. **b** RDF curves of Li-Cl coordination in electrolytes with different C-2Cl diluents. **c** Percentages of different C-2Cl structures in different electrolytes obtained by MD simulations. We define the error bar as the difference between the maximum and minimum in the three data statistics. **d** Δδ of C-Cl characteristic peaks in different electrolytes obtained by FTIR. **e** Δδ of Li⁺ characteristic peaks in different electrolytes obtain by [7]Li NMR. Source data are provided as a Source Data file.

forming stable EEIs on electrodes to improve the electrochemical performance of LMBs[31–33]. Most recently, weakly polar symmetric dichloroalkane (C-2Cl) solvents such as 1,2-dichloroethane (C2-2Cl)[34] have been reported as a diluent candidate in LHCE to enable high stable LMBs, which are expected to satisfy above two criteria as they can regulate electron cloud density via electronegative Cl atoms to weakly coordinate with Li⁺ and confer good flame retardancy by Cl element. However, due to the low boiling point (84 °C) and high melting point (-35 °C) of C2-2Cl diluent, it is difficult to meet the demands of wide temperature range electrolytes. As illustrated in Fig. 1b, a family of symmetric C-2Cl diluents with different carbon chain length (dichloromethane (C1-2Cl), C2-2Cl, 1,3-dichloropropane (C3-2Cl), 1,4-dichlorobutane (C4-2Cl), and 1,5-dichloropentane (C5-2Cl)) were identified as prime candidates, by comparing the melting and boiling points of the different C-2Cl diluents and the frequently utilized TTE diluent in the literature (Fig. 1c and Supplementary Table 1), it becomes evident that as the carbon chain length of the C-2Cl increases, the solvent's liquid temperature range (TR) widens. The C-2Cl solvents with a carbon chain length of≥ 3 exhibit promising potential for wide-temperature-range applications, with TR exceeding that of the TTE diluent. Moreover, these C-2Cl solvents also boast low density and low market price (Fig. 1d), demonstrating the advantages of large-scale application. Nevertheless, the effects of C-2Cl diluents with different

chain lengths on physicochemical properties, solvation structure and electrochemical properties of electrolytes remain unknown.

Based on the above research, here the high-concentration phosphate ester-based electrolyte (LiFSI-TEP with a molar ratio of 1:1.5) was employed as the foundational electrolyte[16,35], and a series of flame-retardant electrolytes were prepared by utilizing C-2Cl with varying carbon chain lengths as diluents in this LiFSI-TEP electrolyte. A systematic investigation into the effects of C-2Cl diluents with diverse carbon chain lengths on the Li⁺ solvation structure, redox behavior, and Li metal interfacial chemistry in the electrolyte was conducted. It is found that as the carbon chain length of the diluent increases, the coordination interaction between the diluent and Li⁺ intensifies. When the carbon chain length ≤ 3, the diluent tends to form a stable cyclic chelating structure with Li⁺, thereby achieving a good electrochemical stability of the diluent. By comprehensively evaluating the physicochemical properties and electrochemical inertness of the diluent, C3-2Cl is identified as the ideal diluent with a wide temperature range (−99 to +120 °C). Based on the theoretical calculation and experimental results, the weak coordination interaction between C3-2Cl and Li⁺ is substantiated, which not only effectively enhances the ionic transport and de-solvation characteristics of the electrolyte across a wide temperature range, but also ensures anion-derived inorganic-rich EEIs on both positive electrode and negative electrode, thereby significantly

improving the cycling stability, rate capability, and low-temperature performance of LMBs. Employing the optimized flame-retardant electrolyte (LiFSI-TEP/C3-2Cl with molar ratio of 1:1.5:3), a Li (50 μm)||LiNi$_{0.5}$Co$_{0.2}$Mn$_{0.3}$O$_2$ (NCM523, 2.0 mAh cm$^{-2}$) coin cell achieves stable cycling for 200 cycles under a high cut-off voltage of 4.5 V (capacity retention rate of 81.7%), and a practical Li (50 μm)|| LiNi$_{0.83}$Co$_{0.12}$Mn$_{0.05}$O$_2$ (NCM83, 5.6 mAh cm$^{-2}$) pouch cell also exhibits stable cycling performance over 100 cycles with a high capacity retention rate of 94.1%. This type of pouch cell also demonstrates good cycle stability at −20 °C, and delivers high discharge capability within a broad temperature range of −60 to +60 °C.

## Results

### Solvation structures of electrolytes with different diluents

To investigate the influence of varying carbon chain lengths on the electron density distribution of C-2Cl diluents, density functional theory (DFT) calculations were initially employed to compare the electrostatic potential (ESP) distributions of these C-2Cl molecules (Fig. 2a and Supplementary Data 1). Electrons are predominantly concentrated around the Cl atom due to its strong electronegativity, exhibiting higher negative potentials. Concurrently, with the increase in chain length within the C-2Cl diluents, the minimum ESP (ESP$_{min}$) becomes lower except for C1-2Cl (partial electron clouds of the Cl atoms are superimposed) (Supplementary Fig. 1), likely due to the increased spacing between 2Cl atoms in longer-chain C-2Cl diluents, which weakens the electron-attracting effects between them. The differences in electron density and molecular structure arising from varying chain lengths lead to distinct coordination interactions and structures between C-2Cl diluents and Li$^+$ (Fig. 2a). When the carbon chain length is ≤ 3, the shorter distance between the 2Cl atoms results in a more concentrated negative potential, prompting the 2Cl atoms in the diluent to synchronously coordinate with a single Li$^+$ more favorably, thereby forming a cyclic chelating structure. Conversely, when the carbon chain length exceeds 3, the interaction between the 2Cl atoms diminishes, favoring the individual Cl atom to coordinate with Li+, potentially leading to structures where a double Li+ is coordinated with a single diluent.

To validate the existence of these structures in the electrolyte, molecular dynamics (MD) simulations, Fourier transform infrared spectroscopy (FTIR), and $^7$Li nuclear magnetic resonance (NMR) were performed to systematically analyze the solvation structures of electrolytes containing different diluents. The molar ratios of LiFSI salt, TEP solvent, and C-2Cl diluents were adjusted to 1:1.5:2. The optimized MD simulations snapshots (Supplementary Fig. 2 and Supplementary Data 1) and the resulting radial distribution function (RDF) curves (Supplementary Fig. 3) reveal that the primary solvation shell of Li$^+$ (r ≈ 2.0 Å) in these electrolytes is predominantly composed of FSI$^-$ anions and TEP solvents. As expected, with the increase in carbon chain length of the C-2Cl diluents, more diluents participate in the Li$^+$ solvation structure, evidenced by the increasing peak intensity of the Li-Cl coordination (r ≈ 2.4 Å) in the RDF curves as the diluent chain length increases (Fig. 2b). According to the MD simulation results, the coordination ratios of Li$^+$ with C-2Cl diluents in different electrolytes are further quantified (Fig. 2c). When the carbon chain length is ≤ 3, single Li$^+$ coordinated C-2Cl diluent structures (SLCC) appear in the electrolytes, with their proportion increasing as the diluent chain length increases. When the carbon chain length exceeds 3, not only are higher contents of SLCC structures present in the electrolytes, but double Li$^+$ coordinated C-2Cl diluent structures (DLCC) are also observed. These findings align well with the DFT calculations, validating the differential impact of varying chain lengths of C-2Cl diluents on the solvation structures of electrolytes.

Additionally, FTIR spectra (Supplementary Fig. 4) reveal that the characteristic peaks of the C-Cl functional groups in C-2Cl diluents appear between 700 - 750 cm$^{-1}$[36,37], with these peaks experiencing a

certain degree of blue-shift in the electrolytes, where the magnitude of the peak-shift (Δδ) reflects the strength of the coordination interaction between the diluent and Li$^+$. As shown in Fig. 2d, by comparing the shifts in the FTIR characteristic peaks of C-2Cl diluents in different electrolytes, it can be found that Δδ gradually increases with the growth of the diluent carbon chain length, further validating that the coordination interaction between the diluent and Li$^+$ strengthens with increasing diluent chain length. $^7$Li NMR was also employed to study the solvation structures of Li$^+$ in different electrolytes (Supplementary Fig. 5), where the characteristic peak positions of Li$^+$ shift from the low-field region to the high-field region upon the addition of diluents, due to the shielding effect caused by the coordination of Li$^+$ and the C-2Cl diluents. Interestingly, Fig. 2e shows that Δδ of $^7$Li NMR initially increases and then decreases with the growth of the diluent carbon chain length. In conjunction with the analysis of the coordination structures between the diluent and Li$^+$ discussed above, we can infer that the short-chain (carbon chain length ≤ 3) C-2Cl diluents are more prone to forming cyclic chelating structures with Li$^+$, which exhibit a stronger shielding effect on Li$^+$, resulting in a larger Δδ. When the diluent carbon chain length increases from 3 to 4, Δδ significantly decreases, validating the weakening of the chelating interaction between the diluent and Li$^+$ when the carbon chain length exceeds 3. The different coordination structures between the solvent and Li$^+$ also influence the binding energy (E$_B$) between them, thereby affecting the desolvation capability of Li$^+$. The E$_B$ of different coordination structures in the electrolytes was calculated by DFT (Supplementary Fig. 6 and Supplementary Data 1), showing the increasement of the E$_B$ with increasing carbon chain length of diluent in the Li-C-2Cl structure, and higher E$_B$ was obtained in the Li-C-2Cl# chelating structures formed by the short-chain diluents with Li$^+$. Besides, all diluents exhibit significantly lower E$_B$ with Li$^+$ than those of FSI$^-$ or TEP with Li$^+$, indicating weak coordination capability of the C-2Cl diluents.

The influences of the coordination between diluents and Li$^+$ on the Li salt solubility and the ionic conductivity of electrolytes were investigated. Firstly, 0.1 M LiFSI salt was added in different pure C-2Cl diluents for 12 h stirring, then the Li solubility was obtained and quantified by inductively coupled plasma optical emission spectrometer (ICP-OES) (Supplementary Fig. 7). Even at this ultra-low Li concentration of 0.1 M, a large amount of undissolved Li salt can still be clearly observed at the bottom. ICP-OES results demonstrate that the Li salt solubility in pure diluent increases with the chain length of C-2Cl diluent, which is consistent with the above analysis of electrolyte solvation structures, as more diluents participate in the Li+ solvation structure with the increase in carbon chain length of the C-2Cl diluents. These results indicate that it is difficult to effectively dissociate sufficient Li salts in pure C-2Cl diluents, so C-2Cl diluents need to interact with TEP solvent to achieve electrolytes with effective ion transport properties. Through testing the viscosity and ionic conductivity of electrolytes containing different C-2Cl diluents, it is found that the similar viscosity is achieved in these electrolytes, but the ionic conductivity slightly increases first and then decreases with the increase of chain length (Supplementary Fig. 8). This may be affected by a combination of complex factors such as the solvation energy of diluents on Li$^+$ and the volume of the Li$^+$ solvation sheath[38]. The stronger solvation energy of long-chain diluents with Li$^+$ helps to improve the ionic conductivity, but the larger volume of Li$^+$ solvation sheath is not conducive to Li$^+$ transport and thus reduces ion conductivity.

### Redox behavior of electrolytes with different diluents

The stability of the electrolytes with different diluents towards Li metal can be evaluated by the plating/stripping Coulombic efficiency of Li|| Cu coin cells. Under the testing conditions of a current density of 0.1 mA cm$^{-2}$ and a capacity of 1 mAh cm$^{-2}$, the initial Coulombic efficiency (ICE) of Li||Cu cells in different electrolytes (Fig. 3a) and the magnitude of polarization voltage (Fig. 3b) were compared. Compare

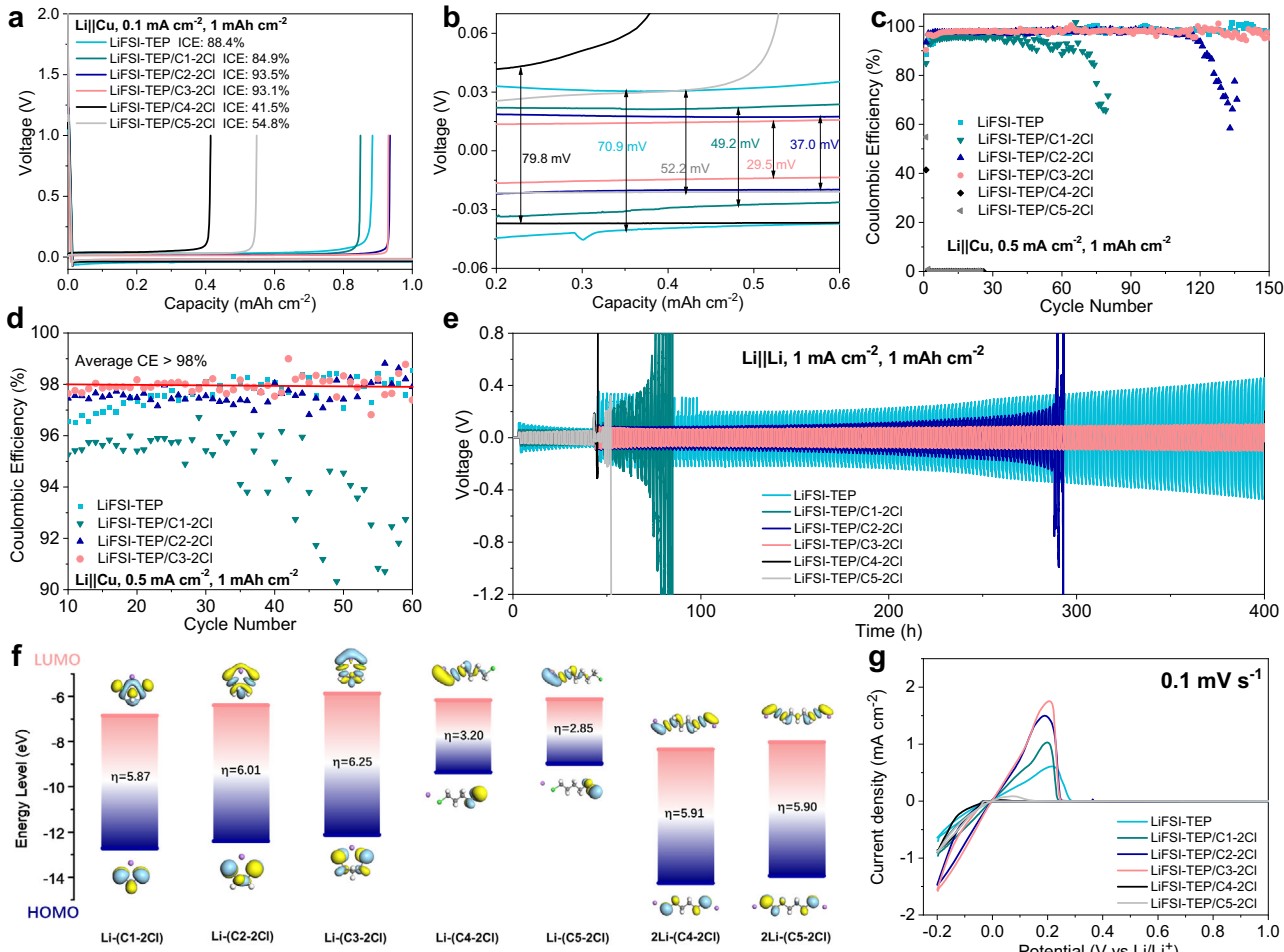

**Fig. 3 | Compatibility of Li metal in electrolytes with different C-2Cl diluents.** **a** ICE of Li||Cu cells in different electrolytes. **b** Initial polarization of Li||Cu cells in different electrolytes. **c** Cycling performance of Li||Cu cells in different electrolytes under conditions of 0.5 mA cm⁻² and 1 mAh cm⁻² and **d** Corresponding average CE of Li||Cu cells from 10th to 60th cycles. **e** Long-term Li plating/stripping performance of Li||Li symmetric cells in different electrolytes. **f** HOMO/LUMO energy levels of different Li-C-2Cl coordination complexes obtained by DFT calculations. Color code: Li-purple, C-dark gray, H-white, Cl-green. **g** CV curves of Li||Cu cells with different electrolytes under a scanning rate of 0.1 mV s⁻¹. Source data are provided as a Source Data file.

to the ICE of Li||Cu cell with the LiFSI-TEP electrolyte (88.4%), the electrolytes with long-chain diluents significantly reduced the ICE (41.5% in C4-2Cl and 54.8% in C5-2Cl), whereas those electrolytes with short-chain diluents still exhibited high ICE, with improvements in ICE observed in C2-2Cl (93.5%) and C3-2Cl (93.1%) diluents. Simultaneously, electrolytes with short-chain diluents also demonstrated smaller polarization voltages in Li||Cu cells (Fig. 3b). These results indicate that short-chain diluents not only effectively reduce the viscosity of the electrolyte but also provide good stability to the Li metal and suppress interfacial side reactions, thereby decreasing battery polarization and enhancing Li metal plating/stripping ICE. By comparing the cycling performance of Li||Cu cells at a current density of 0.5 mA cm⁻² and a capacity of 1 mAh cm⁻² (Fig. 3c), it is observed that cells in electrolytes with long-chain diluents exhibited rapid performance degradation early on, whereas those with short-chain diluents can achieve stable cycling performance, with the longest cycle life over 150 cycles in the electrolyte with C3-2Cl diluent. Notably, at the initial 60 cycles, the average CEs of Li||Cu cells in different electrolytes were compared, showing the highest average CE (> 98%) of Li||Cu cell in the electrolyte with C3-2Cl diluent (Fig. 3d).

Furthermore, more comprehensive CE tests in Cu||NCM523 (areal loading of 2.0 mAh cm⁻²) full cells with limited Li inventory were performed (Supplementary Fig. 9). As expected, at the initial 0.05 C charge-discharge process, the Cu||NCM523 full cell with LiFSI-TEP/C3-

2Cl exhibits the highest ICE (83.3%) and the highest initial discharge capacity (170.9 mAh g⁻¹). Meanwhile, it also displays the most stable cycling performance with the highest capacities and the highest CEs (> 97.4%) at 0.1 °C, indicating that the C3-2Cl diluent has the best electrochemical stability compared to other C-2Cl diluents. Additionally, Li||Li symmetric cells were also assembled to compare the Li plating/stripping stability of different electrolytes with Li metal, which were cycled at a high current density of 1 mA cm⁻² and a capacity of 1 mAh cm⁻² after 20 cycles activation at a low current density of 0.25 mA cm⁻² and a capacity of 0.25 mAh cm⁻² (Fig. 3e). Li||Li symmetric cells exhibit similar results with Li||Cu cells, showing that the electrolytes with long-chain diluents exhibit rapid performance failure at initial cycles after activation process, whereas the electrolytes with short-chain diluents maintain prolonged cycles. Notably, the Li||Li symmetric cell using the electrolyte with C3-2Cl diluent exhibits the lowest polarization voltage and stable Li plating/stripping performance for over 400 h, which is much better than that with LiFSI-TEP electrolyte.

In addition, electrochemical impedance spectroscopy (EIS) was used to measure the impedance of Li||Li symmetric cells with different electrolytes after 50 h cycles (Supplementary Fig. 10). It is found that the batteries in electrolytes with long-chain diluents exhibit significant increases in impedance after cycling, indicating severe electrolyte consumption and interfacial deterioration due to serious interfacial

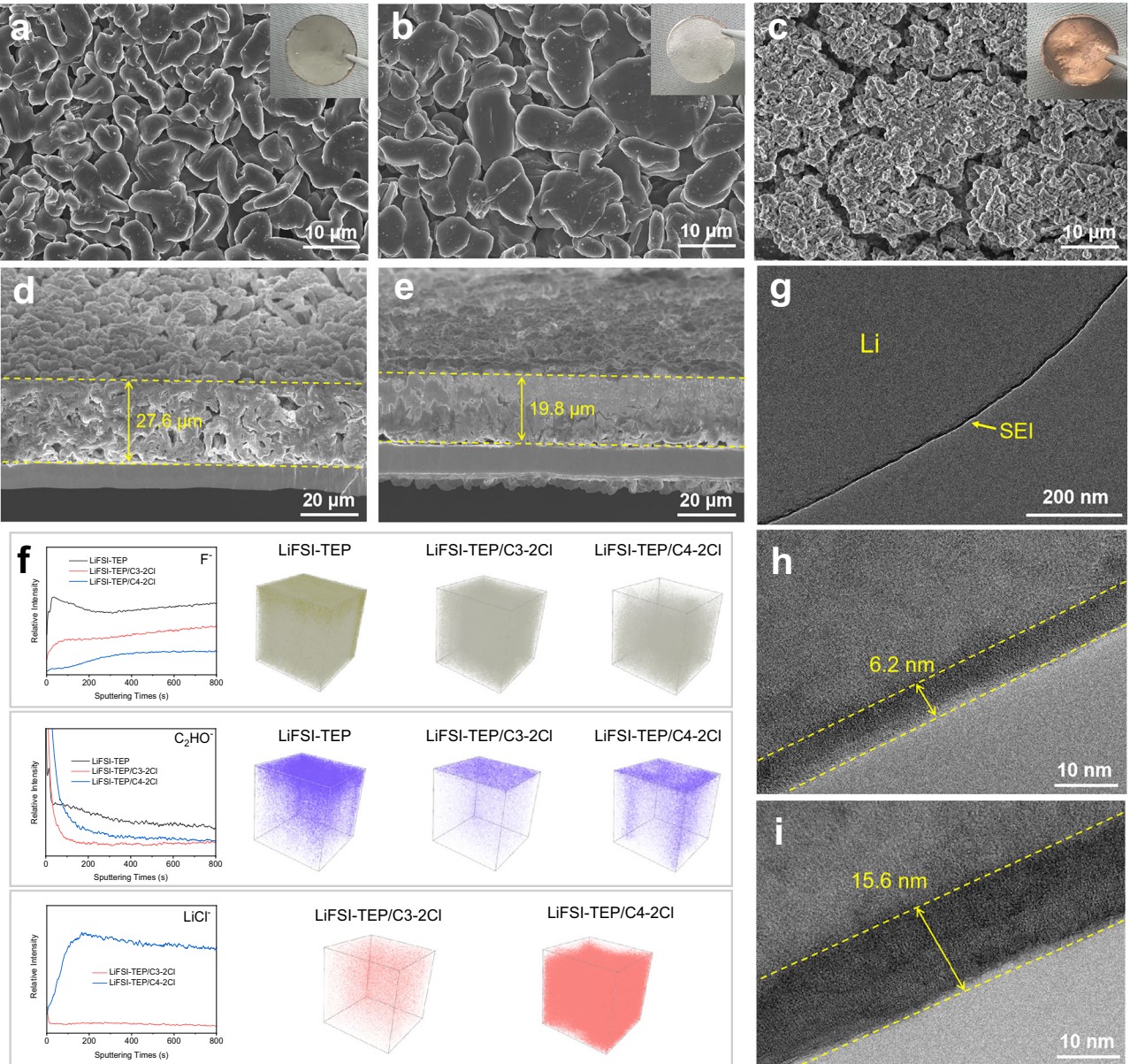

**Fig. 4 | Li deposition morphology and interfacial chemistry of Li metal electrodes in different electrolytes.** Top-view SEM images of the surface morphology of Li plated on Cu foil with a current density of 0.5 mA cm$^{-2}$ and an areal capacity of 3 mAh cm$^{-2}$ in **a** LiFSI-TEP, **b** LiFSI-TEP/C3-2Cl, **c** LiFSI-TEP/C4-2Cl. The insets are optical images of the corresponding Li-plated Cu foils with a diameter of 1.6 cm. Cross-sectional SEM images of Li plated on Cu foil with a current density of 0.5 mA cm$^{-2}$ and an areal capacity of 3 mAh cm$^{-2}$ in **d** LiFSI-TEP, **e** LiFSI-TEP/C3-2Cl. **f** TOF-SIMS depth profiles and distributions of F$^-$, C$_2$HO$^-$, LiCl$^-$ species on the surface of Li metal electrode after 50 cycles in different electrolytes. Cryo-TEM characterization of the morphology and thickness of the SEI layers formed on Li metal surface in **g, h** LiFSI-TEP/C3-2Cl and **i** LiFSI-TEP. Source data are provided as a Source Data file.

side reactions, leading to battery failure. In contrast, batteries in the electrolytes with short-chain diluents show significantly reduced impedance compared to that in LiFSI-TEP electrolyte, in which the electrolyte with C3-2Cl diluent exhibits the smallest interfacial impedance, suggesting the formation of a low-impedance and stable solid electrolyte interphase (SEI) on the Li metal surface, thereby effectively improving the electrochemical performance of the LMBs. Additionally, coulometric titration time analysis (CTTA) with Li||Cu cell was performed to quantify the side reactions between electrolyte and Li metal electrode[39]. The potential profiles of the various electrolytes during the 400 h test and the total accumulated charge consumed in side reactions (q$_\Sigma$) were shown in Supplementary Fig. 11. The order of accumulated charge consumed after 400 h was: LiFSI-TEP/C4-2Cl > LiFSI-TEP/C5-2Cl > LiFSI-TEP/C1-2Cl > LiFSI-TEP/C2-2Cl > LiFSI-TEP >

LiFSI-TEP/C3-2Cl. This indicates that C4-2Cl and C5-2Cl exhibit the highest reactivity with Li metal electrode, while C3-2Cl demonstrates the highest stability among all symmetric dichloroalkane diluents. To verify the above conclusion, the Li||Cu cells were also assembled for potentiostatic holding tests under the conditions of 0.1 V (eliminating the interference caused by Li reduction) for 50 h. As shown in Supplementary Fig. 12, after 50 h of potentiostatic holding, the accumulated charge of the Li||Cu cell with LiFSI-TEP/C3-2Cl is 16.9 μAh, which is lower than that of TEP-based electrolytes with other C-2Cl diluents and the LiFSI-TEP electrolyte, verifying the highest electrochemical stability of the C3-2Cl diluent.

All the above results demonstrate that the diluents with different carbon-chain lengths significantly affect the stability of electrolytes and Li metal. In order to figure out the reason, we first investigated the

chemical reactions between the different diluents and Li metal. Fresh Li metal sheets were separately soaked in different pure diluents for 2 weeks. Supplementary Fig. 13 obviously displays that after soaking Li metal in five pure diluents for 2 weeks, all the diluents still remain transparent and the surfaces of the five Li metal sheets still present bright metallic luster. The soaking experiments indicate that these diluents from C1-2Cl to C5-2Cl are chemically stable with Li metal. We then investigated the electrochemical redox behavior of the electrolytes with different diluents using DFT calculations. The highest occupied molecular orbital (HOMO) and lowest unoccupied molecular orbital (LUMO) energies of different Li$^+$-diluent complexes were obtained by DFT calculations. As shown in Fig. 3f, short-chain C-2Cl diluents forming chelating structures with a single Li$^+$ exhibited lower HOMO energies compared to long-chain (carbon chain length > 3) diluents coordinating with a single Li$^+$, thereby the electrolytes with short-chain diluents showed better oxidation resistance. Simultaneously, due to the electrolytes with long-chain diluents tending to form the DLCC structures, their LUMO energies are significantly decreased; hence, the electrolytes with long-chain diluents exhibit poorer reduction stability. It is worth noting that the chelating structure of hexatomic Li-C3-2Cl complex shows a higher LUMO energy level than that of pentatomic Li-C2-2Cl complex or tetratomic Li-C1-2Cl complex, indicating a higher stability of the hexatomic chelating solvation structure and higher reduction stability of the electrolyte with C3-2Cl diluent. The electrochemical inertness of the compound can also be identified by the difference value of HOMO and LUMO energies (η), showing that the SLCC structure of Li-C3-2Cl has the best electrochemical inertness due to its highest η value (6.25). Therefore, the electrolyte with C3-2Cl diluent achieved the highest average Coulombic efficiency and longest cycle life in LMBs among these diluents.

To further investigate the electrochemical oxidation behavior of the electrolytes with different diluents, Linear sweep voltammetry (LSV) of Li||Al cells with different electrolytes was performed. As shown in Supplementary Fig. 14, LSV curves of Li||Al cells with different electrolytes indicate that the oxidation potentials of electrolytes containing short-chain diluents exhibit a better oxidation stability beyond 5 V (vs. Li/Li$^+$), suggesting their potential application in high-voltage battery systems. Cyclic voltammetry (CV) of Li||Cu cells was also measured to evaluate the electrochemical reduction behavior of electrolytes with different diluents. From Fig. 3g, it is observed that Li metal exhibits firstly increasing plating/stripping reversibility in electrolytes containing short-chain diluents from C1-2Cl, C2-2Cl, to C3-2Cl. However, when the carbon chain length of the diluents increases to C4-2Cl and C5-2Cl, Li metal reversible plating/stripping behavior gets worse, even much worse than LiFSI-TEP electrolyte. This phenomenon is consistent with the above results, demonstrating that the electrolytes with C1-2Cl, C2-2Cl, and C3-2Cl diluents show good electrochemical reduction stability, but the electrolytes with C4-2Cl or C5-2Cl diluents exhibit bad electrochemical reduction stability. The enlarged CV curves from 2.0 V to 0 V in the reduction process display that typical reduction peaks at -1.2 V ascribed to LiFSI decomposition (Supplementary Fig. 15)[40]. No other significant reduction peak observed in the electrolytes with C4-2Cl or C5-2Cl diluents may be due to the electrochemical reduction peak overlaps with Li deposition process.

## Li deposition morphology and SEI chemistry on the Li metal surface

The interfacial chemistry between the electrolyte and Li metal electrode significantly influences the surface morphology of the Li metal electrode and the composition of the SEI layer, thereby affecting battery performance. To investigate the impact of different diluents on the deposition behavior of Li metal, scanning electron microscopy (SEM) was employed to observe the surface morphology of Li metal electrodes after 50 cycles in different electrolytes. In the LiFSI-TEP electrolyte without diluent, the cycled Li metal surface exhibited a

loose and porous morphology with varying moss-like particles (Supplementary Fig. 16a). By contrast, the similar morphological structure was also observed on the Li metal surface cycled in the electrolyte with C1-2Cl diluent (Supplementary Fig. 16b), while the C2-2Cl and C3-2Cl groups predominantly showed larger and more uniform deposition morphology (Supplementary Fig. 16c, d). In sharp contrast, severe cracking and pulverization phenomena were observed on the Li metal surface cycled in electrolytes with long-chain diluents (Supplementary Fig. 16e, f). Similarly, the top-view and cross-sectional morphology of Li metal plated on Cu foil with a current density of 0.5 mA cm$^{-2}$ and an areal capacity of 3 mAh cm$^{-2}$ was also observed by SEM. Compared with the loose, porous and thick (~27.6 μm) deposition morphology of Li metal in LiFSI-TEP (Fig. 4a, d), the deposition morphology of Li metal in LiFSI-TEP/C3-2Cl was larger and denser (Fig. 4b), and the thickness was thinner (~19.8 μm) (Fig. 4e). Meanwhile, the Li deposited in LiFSI-TEP/C3-2Cl also showed a brighter metallic luster than that deposited in LiFSI-TEP (insets in Fig. 4a and b). However, in the LiFSI-TEP/C4-2Cl, only a small amount of Li metal was deposited on the surface of the Cu foil, and it showed a black and loose state like dead Li (Fig. 4c and inset). Upon disassembling the batteries after cycling in the electrolytes with long-chain diluents, significant drying of the electrolyte on the internal separators were evident (Supplementary Fig. 17), indicating severe side reactions between the electrolytes and the Li metal. These phenomena are also consistent with the large interfacial impedance and rapid battery failure exhibited by the LMBs discussed above.

To further analyze the interfacial reactions between different electrolytes and the Li metal electrode, X-ray photoelectron spectroscopy (XPS) was performed to test the composition of the SEI layer on the surface of the cycled Li metal electrode. In F 1$s$ spectra (Supplementary Fig. 18), it is found that the Li metal electrode surfaces cycled in the aforementioned electrolytes all exhibited a high content of LiF formed by the reduction of FSI$^-$ anions. However, through the C 1$s$ (Supplementary Fig. 19) and Cl 2$p$ spectra (Supplementary Fig. 20), it is observed that the Li metal surfaces cycled in electrolytes with long-chain diluents showed a higher content of chemical components such as C-Cl, C-C/C-H, and Li-Cl derived from diluent decomposition, further validating the exacerbation of interfacial side reactions between long-chain diluents and the Li metal. Notably, the Li metal surface cycled in the electrolyte with C3-2Cl diluent shows the lowest content of Cl-containing components derived from diluent decomposition among these five C-2Cl diluents, as confirmed by the lowest atomic ratio of Cl obtained by XPS analysis (Supplementary Fig. 21), demonstrating a higher reduction stability of the electrolyte with C3-2Cl diluent. Furthermore, Time-of-flight secondary ion mass spectrometry (TOF-SIMS) was performed to identify the chemical composition of SEI layer at different depths. As shown in Fig. 4f, the inner layer of SEI formed in the LiFSI-TEP contained high content of F$^-$ and C$_2$HO$^-$, which is mainly formed by the reduction of LiFSI salt and TEP solvent. However, after the addition of diluent, the contents of these two components were decreased, and C3-2Cl had a higher content of F$^-$ and a lower content of C$_2$HO$^-$ compared with C4-2Cl, while the content of LiCl$^-$ was also significantly lower, indicating that long-chain diluent has a high reactivity with Li metal to generate more C and Cl-contained components.

In conjunction with the above results and theoretical calculations discussed above, the possible mechanism of the SEI chemistry on Li metal surface in electrolytes with different chain length C-2Cl diluents is shown in Supplementary Fig. 22. It can be concluded that the long-chain diluent readily forms a high-valent 2Li$^+$-diluent coordination complex with low LUMO energy in the Li$^+$ solvation structure, which is prone to a reduction reaction with Li$^+$ to form dead Li, LiCl and organic species on Li metal surface, thereby causes low Coulombic efficiency and failure of LMBs due to the persistent interfacial side reactions and electrolyte depletion. In contrast, the short-chain diluent benefits to form a low-valent 1Li$^+$-diluent coordination complex with high LUMO

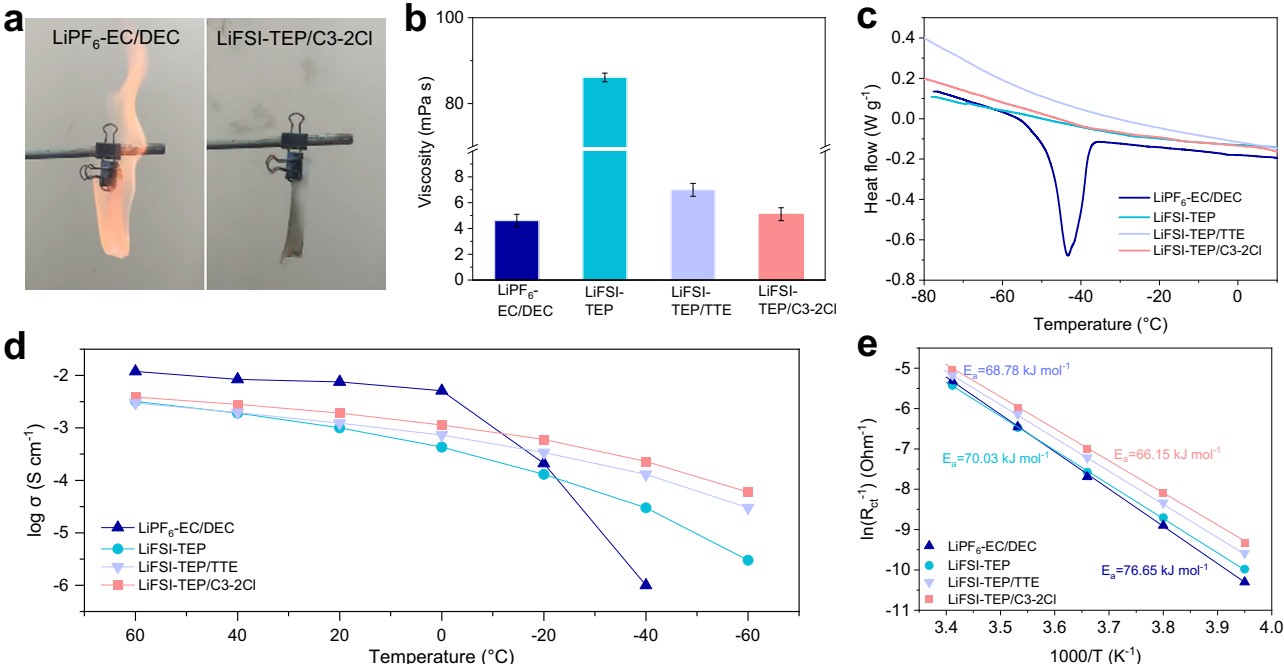

**Fig. 5 | Physicochemical properties of different electrolytes. a** Combustion tests of flammable cotton tapes soaked in different electrolytes. **b** Viscosity of different electrolytes. We define the error bar as the difference between the maximum and minimum in the five replicated measurements. **c** DSC curves of different electrolytes at a freezing rate of 10 °C min⁻¹. **d** Ionic conductivity of different electrolytes over a wide temperature range of −60 to +60 °C. **e** Li⁺ de-solvation activation energy tests of different electrolytes. Source data are provided as a Source Data file.

energy, especially the C3-2Cl diluent readily forms a more stable hexatomic chelating structure with Li⁺, which possesses better reduction stability than pentatomic and tetratomic chelating structures formed by C2-2Cl and C1-2Cl diluents, respectively. Meanwhile, the anion-dominated Li⁺ solvation structure formed in the electrolyte with C3-2Cl diluent can form a robust anion-derived inorganic-rich SEI layer on the Li metal electrode surface, effectively reducing the interfacial side reactions in LMBs and significantly enhancing their electrochemical performance. The SEI layer formed on the Li metal surface was further observed by cryogenic transmission electron microscopy (cryo-TEM), revealing that the SEI layer formed by the LiFSI-TEP/C3-2Cl on the Li metal surface was thin and uniform (Fig. 4g), with an average thickness of ~6.2 nm (Fig. 4h). By contrast, the average thickness of the SEI layer formed by the LiFSI-TEP was ~15.6 nm (Fig. 4i), which is much larger than that in the LiFSI-TEP/C3-2Cl. The thin and homogeneous SEI layer plays a significant role in reducing interfacial impedance, improving Li plating/stripping reversibility and kinetics, and suppressing dendritic Li growth.

In addition to analyzing the stability of the flame-retardant electrolyte on the Li metal electrode, we also tried to apply it to the commercial Li-ion battery system with graphite (Gr) as the negative electrode. First, we have assembled a Gr||NCM523 pouch cell with Gr negative electrode, using the LiFSI-TEP/C3-2Cl electrolyte with a charge-discharge rate of 0.05 C at 25 °C. Regretfully, this pouch cell showed a low discharge capacity at the initial cycle (Supplementary Fig. 23a), indicating some undesired reactions are occurred in this system. Then, the cycled pouch cell was disassembled, showing obvious black reaction mottling and powder dropping phenomena on the surface of Gr electrode (Supplementary Fig. 23b). The X-ray diffraction (XRD) spectra demonstrate that the intensity of typical Gr diffraction peak (002) at 26.5° significantly decreases after first cycle (Supplementary Fig. 23c), suggesting the structure of Gr is partially destroyed by the co-intercalation or reaction of the electrolyte. To further verify the above conjecture, morphologies of the pristine and cycled Gr electrodes were observed by SEM. It is found that a compact

and flat Gr particle was observed in pristine electrode (Supplementary Fig. 23d), but it turns to an exfoliated layered structure after cycling (Supplementary Fig. 23e). The above results prove that this kind of electrolyte is not suitable for Li-ion battery system with Gr negative electrode at this stage, and further research on its reaction mechanism in this system is needed in the follow-up work.

## Physicochemical properties of different electrolytes over a wide-temperature range

Above results indicate that the C3-2Cl diluent is more conducive to the stability of the electrolyte with respect to the Li metal electrode, and considering its wide liquid temperature range (-99 to +120 °C), thereby it was selected as the preferred diluent for the electrolyte in wide-temperature LMBs. Initially, the addition ratio of the C3-2Cl diluent in the electrolyte was optimized based on the ionic conductivity and viscosity tests (Supplementary Fig. 24). It is observed that the viscosity of the electrolytes decreased with increasing diluent content, while the room-temperature ionic conductivity initially increased and then decreased, reaching a maximum of 1.91 mS cm⁻¹ at a molar ratio of 1:1.5:3 in LiFSI-TEP/C3-2Cl. Therefore, this ratio was chosen as the preferred formulation for the electrolyte. Furthermore, the combustion characteristic of different electrolytes was measured (Fig. 5a). An intense and sustained combustion was obtained in commercial carbonate-based electrolyte (1 M LiPF₆ in ethylene carbonate (EC)/diethyl carbonate (DEC) with a volume ratio of 1:1) when it was exposed to an open flame (Supplementary Movie 1). In contrast, LiFSI-TEP/C3-2Cl demonstrated good flame retardancy, as combustible cotton tape soaked in this electrolyte could not be ignited (Supplementary Movie 2). Besides, the flammability of pure diluents with different chain-length was also measured in Supplementary Fig. 25, showing that the flammability of the diluents increases with the increasement of C/Cl ratio. As expected, when the carbon chain-length ≤ 3, the diluents can exhibit rapid self-extinguishing characteristics, verifying the flame retardancy of short-chain C-2Cl diluents. On the other hand, the viscosity of the electrolyte was significantly reduced after C3-2Cl addition, which improved the

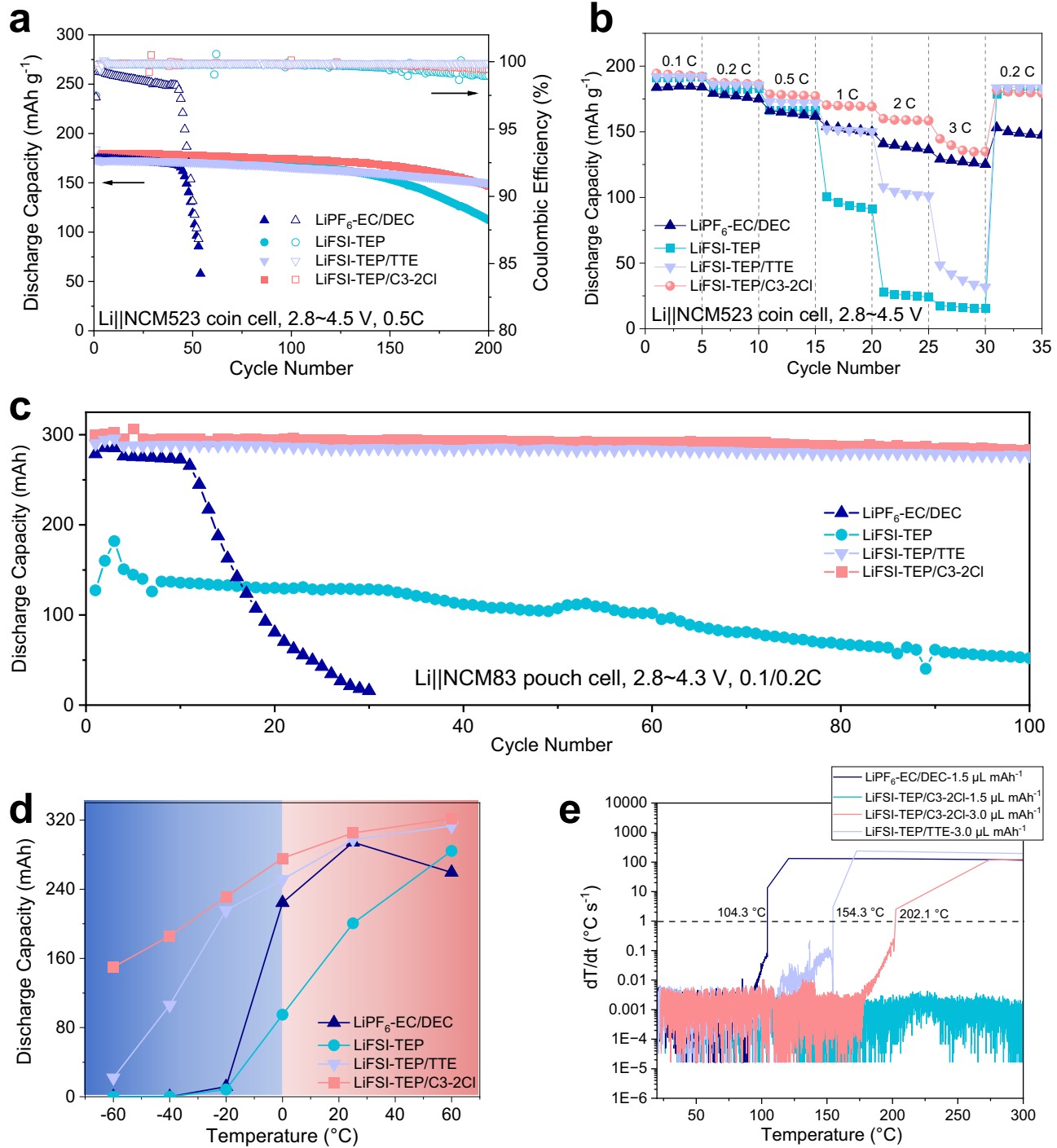

**Fig. 6 | Electrochemical performance of high-voltage LMBs with different electrolytes. a** Cycling performance of Li||NCM523 coin cells with different electrolytes under test conditions of 2.8 - 4.5 V, 0.5 C charge/discharge and 25 °C. **b** Rate capability of Li||NCM523 coin cells with different electrolytes. **c** Cycling performance of Li||NCM83 pouch cells with different electrolytes under test conditions of 2.8 - 4.3 V, 0.1 C charge/0.2 C discharge and 25 °C. Inset is an optical image of the pouch cell. **d** Discharge capacities of Li||NCM83 pouch cells with different electrolytes under different temperatures. **e** ARC tests of Li||NCM83 pouch cells with different electrolytes and different electrolyte amounts. Source data are provided as a Source Data file.

wettability to polypropylene (PP) separator (Fig. 5b and Supplementary Fig. 26). The optimized electrolyte's viscosity was comparable to that of a commercial carbonate-based electrolyte (5.1 vs. 4.6 mPa s), and its contact angle on the separator was superior (33° vs. 46°). Additionally, the state-of-the-art TTE diluent was also employed in LiFSI-TEP electrolyte (molar ratio of 1:1.5:3 in LiFSI-TEP/TTE) for comparison, which exhibits a slightly higher viscosity (7.1 mPa s) but similar contact angle (33°) due to the high surface tension of fluorine-containing TTE diluent.

Furthermore, the Li$^+$ transference numbers of the different electrolytes were measured to identify the Li$^+$ transport property in bulk electrolyte (Supplementary Fig. 27). The addition of the C3-2Cl diluent increased the Li$^+$ transference number of the LiFSI-TEP electrolyte from 0.62 to 0.74, far exceeding the value (0.32) of commercial carbonate-based electrolyte and also higher than the value (0.68) of state-of-the-art LiFSI-TEP/TTE, indicating that the C3-2Cl diluent is beneficial for enhancing the Li$^+$ transport property in the electrolyte.

To study the potential application of the electrolyte in a wide temperature range, differential scanning calorimetry (DSC) was further used to test the freezing points of the three different electrolytes at low temperatures. During the cooling process, the commercial carbonate-based electrolyte exhibited a distinct exothermic peak at around −40 °C (Fig. 5c), corresponding to the electrolyte's freezing point. By contrast, no freezing point was observed in the other three flame-retardant electrolytes even when cooled to −80 °C, indicating that these electrolytes can remain in a liquid state at low temperatures. Then, the ionic conductivity of these electrolytes over a wide temperature range was measured (Fig. 5d and Supplementary Fig. 28). Although the electrolyte with C3-2Cl diluent has a relative lower ionic conductivity (1.91 mS cm$^{-1}$) than that of the commercial carbonate electrolyte (7.53 mS cm$^{-1}$) at 25 °C, it shows a high ionic conductivity (0.60 mS cm$^{-1}$) at −20 °C, exceeding that of the LiFSI-TEP/TTE (0.34 mS cm$^{-1}$), LiFSI-TEP (0.13 mS cm$^{-1}$) and the commercial carbonate-based electrolyte (0.21 mS cm$^{-1}$). Even at −60 °C, the ionic conductivity of the electrolyte with C3-2Cl diluent was significantly higher than that of the other three electrolyte systems. Furthermore, Li$^+$ de-solvation activation energy (E$_a$) was calculated based on the Arrhenius equation, which was measured by probing the charge transfer resistance (R$_{ct}$) of Li∥Li symmetric cells under different temperatures (Supplementary Fig. 29)[41,42]. By comparing the E$_a$ of these three electrolytes (Fig. 5e), the electrolyte with C3-2Cl diluent also exhibited the lowest E$_a$ value, indicating a faster Li$^+$ de-solvation capability in this electrolyte. The enhancement of Li$^+$ transport and de-solvation capability is mainly due to the weakly coordination ability between Li$^+$ and C3-2Cl diluent, which is different from TTE diluent with minimal coordination ability with Li$^+$ as reported by literatures[27,28]. Consequently, the addition of C3-2Cl diluent can bring numerous benefits to the electrolyte, including low viscosity, a wide liquid temperature range, non-flammability, enhanced Li$^+$ transport, and de-solvation capabilities. The improvement in these electrolyte properties is crucial for realizing the application in wide-temperature LMBs systems.

## Application of electrolytes in practical high-voltage LMBs under extreme conditions

To investigate the potential application of electrolytes in high-voltage LMBs, the stability of Al current collectors in different electrolytes was firstly investigated by using Li∥Al cells with 4.5 V constant voltage charging for 24 h (Supplementary Fig. 30). As reported in literature[19], the electrolyte with low LiFSI salt concentration (1 M LiFSI in TEP, namely LCE) shows severe corrosion to Al foil, which delivers high corrosion current under 4.5 V charging process and causes corrosion pitting on Al foil surface. In contrast, low corrosion currents and no obvious signs of corrosion were observed in high-concentration LiFSI-TEP (namely HCE) and LiFSI-TEP/C3-2Cl electrolytes, indicating a high stability of Al foils and good application potentials of these electrolytes in high-voltage batteries. XPS spectra on the Al foil surface demonstrate that F-containing species derived from LiFSI salt corrosion were detected in LCE, while no obvious F-containing or Cl-containing species were formed in high-concentration LiFSI-TEP with/without C3-2Cl diluent, further verifying the few corrosions between the Al current collector and Cl-containing electrolytes.

Afterward, high-voltage LMBs were assembled using a NCM523 positive electrode (areal loading of 2.0 mAh cm$^{-2}$) and a 50 μm Li metal negative electrode (N/P ratio ~5, electrolyte amount ~25 μL mAh$^{-1}$). Under a high cutoff voltage of 4.5 V and a charge/discharge rate of 0.5 C (1 C = 170 mA g$^{-1}$) at 25 °C, the Li∥NCM523 coin cell employing the commercial carbonate-based electrolyte exhibited a "capacity plunge" begin at around 40th cycle (Fig. 6a), primarily due to the oxidation decomposition of the carbonate-based electrolyte at high voltages and severe interfacial side reactions with the Li metal electrode, leading to electrolyte depletion and interfacial degradation. By contrast, Li∥NCM523 coin cells assembled with flame-retardant

electrolytes demonstrated superior cyclability, with the electrolyte containing C3-2Cl diluent outperforming the LiFSI-TEP electrolyte in terms of capacity utilization and cycle retention, maintaining 81.7% of its capacity after 200 cycles. Meanwhile, the slow-rate (0.05 C) cycling with 1 h waiting time at 4.5 V charged state in Li∥NCM523 cell with LiFSI-TEP/C3-2Cl was also tested (Supplementary Fig. 31), which reveals almost no attenuation capacity within initial 20 cycles, indicating a high stability of the electrolyte in this system. Consistent with the previous report[18,26], the electrolyte with TTE diluent also displayed superior cycling stability and high Coulombic efficiency in Li∥NCM523 cell. However, rate capability tests reveal that the electrolyte with C3-2Cl diluent effectively improved capacity utilization under high-rate charge/discharge conditions (Fig. 6b), which is much better than the LiFSI-TEP with/without TTE diluent. Above results are mainly attributed to the weakly coordination between Li$^+$ and C3-2Cl diluent, which effectively enhances Li$^+$ transport/de-solvation characteristics.

To further validate the potential of electrolytes in practical LMBs, a NCM83 positive electrode with a high loading of 5.6 mAh cm$^{-2}$ and a 50 μm Li metal negative electrode (N/P ratio ~1.8, electrolyte amount ~5 μL mAh$^{-1}$) were utilized, assembling multi-layer pouch cells with capacity exceeding 300 mAh. Under a voltage profile of 2.8 - 4.3 V and a 0.1 C charge/0.2 C discharge rate (1 C = 200 mA g$^{-1}$), the cycling performance of Li∥NCM83 pouch cells was tested at 25 °C (Fig. 6c). Due to the high viscosity and low ionic conductivity of the LiFSI-TEP electrolyte, the Li∥NCM83 pouch cell exhibited limited capacity utilization, retaining only 52.2 mAh after 100 cycles. By contrast, the addition of the C3-2Cl diluent improved the overall performance of the electrolyte, enabling the Li∥NCM83 cell to achieve high capacity utilization and stable cycling for over 100 cycles (capacity retention ~94.1%). Besides, the electrolyte with TTE diluent also exhibits stable cycling performance in Li∥NCM83 pouch cell, which is consistent with the previous study[18]. As a contrast, the pouch cell with the commercial carbonate-based electrolyte showed a rapid capacity decay trend after just 10 cycles, with little capacity retention after 30 cycles, consistent with the cycling performance of the coin cells obtained above.

Then, we disassembled the pouch cells after 50 cycles and SEM was performed to analyze the cycled Li metal electrode morphology, revealing the formation of numerous sharp Li dendrites and porous deposition on the surface of the Li metal electrode after cycling in the commercial carbonate-based electrolyte (Supplementary Fig. 32a). Satisfyingly, the Li metal surface after cycling in the electrolyte containing the C3-2Cl diluent was more planar and denser (Supplementary Fig. 32b), which is beneficial for long-term cycling stability and safety of LMBs. Furthermore, the Top-view (Supplementary Fig. 33) and cross-sectional (Supplementary Fig. 34) morphologies of the NCM83 electrode after 50 cycles in the Li∥NCM83 pouch cell with electrolyte containing the C3-2Cl diluent were also observed by SEM, revealing that the single-crystal structure of the positive electrode material remained intact, with no apparent microcracks observed internally. Meanwhile, TEM analysis reveals that a uniform cathode-electrolyte interphase (CEI) layer with an average thickness of ~4 nm was formed on the NCM83 particle surface (Supplementary Fig. 35). XPS analysis indicates that the CEI layer was primarily composed of compounds formed by the decomposition of FSI$^-$ anions, containing F, N, O, and S, with fewer P and Cl components from the oxidation of TEP solvent and C3-2Cl diluent (Supplementary Fig. 36). To compare the difference in CEI layer at higher cut-off voltage, 4.5 V NCM523 positive electrodes were also disassembled and characterized from coin cells after 50 cycles in TEP-based electrolytes with/without C3-2Cl diluent (Supplementary Fig. 37). Compare to the CEI layer formed by LiFSI-TEP, a thin (~5 nm) and uniform CEI layer was also observed on the surface of 4.5 V NCM523 positive electrode in LiFSI-TEP/C3-2Cl, which consists of negligible Cl-contained species and less C/P-contained species derived by C3-2Cl diluent and TEP solvent. These results indicate that C3-2C1 diluent is also electrochemically inert and can't be decomposed at high

voltage. All in all, the above results fully demonstrate that the electrolyte containing the C3-2Cl diluent can form a thin CEI layer derived from anion decomposition on the positive electrode surface, which effectively inhibits the side reactions between reactive solvents and the electrode, ensuring the stability and integrity of the electrode structure during cycling.

To assess the application capability of different electrolytes over a wide temperature range, the discharge capacity of Li∥NCM83 pouch cells at various temperatures was collected. The cells were charged at 0.1 °C and then discharged at 25 °C, followed by discharge at different temperatures. The results show that due to the slow Li$^+$ transport and de-solvation processes of the commercial carbonate-based and high-concentration LiFSI-TEP electrolytes at low temperatures, the cells struggled to effectively utilize their capacity at −20 °C (Fig. 6d and Supplementary Fig. 38). At 60 °C high temperature, although Li$^+$ transport kinetics is improved, the side reactions between the electrolyte and electrode are also intensified, leading to capacity decay in cells with the commercial carbonate-based electrolyte. Compared to LiFSI-TEP electrolyte, although the low temperature performance of the Li∥NCM83 pouch cell has been improved after the addition of TTE diluent, the low temperature discharge capacity is still insufficient to meet the application requirements. Notably, Li∥NCM83 pouch cell with the electrolyte containing the C3-2Cl diluent demonstrated good potential for wide-temperature applications, operating stably across a temperature range of −60 to +60 °C, with capacity utilization reaching 50% of 25 °C capacity even at a low temperature of −60 °C. Additionally, even charge/discharge processes were both operated at 60 °C high temperature or −60 °C low temperature, Li∥NCM83 pouch cell with the electrolyte containing the C3-2Cl diluent still exhibited stable charge-discharge profiles during several cycles, manifesting the application potential of this electrolyte in wide temperature range (Supplementary Fig. 39).

In order to further investigate the safety performance of the flame-retardant electrolytes, 2 Ah-level Li∥NCM83 pouch cells (N/P ratio ~1.8, electrolyte amount ~1.5 μL mAh$^{-1}$) with practical gravimetric specific energy of 418.8 Wh kg$^{-1}$ were assembled (Supplementary Fig. 40). Accelerated rate calorimetry (ARC) was performed to comprehensively evaluate the safety of pouch cells utilizing different electrolytes[43]. Three key temperatures are crucial for identifying the safety performance of batteries. The onset temperature of self-heating, denoted as T1, marks the beginning of internal chain reactions that lead to an autonomous and sustained increase in temperature. Once the temperature surpasses T1, it becomes unsafe to operate or store the batteries. T2 is identified as the threshold temperature for thermal runaway, defined by a rate of temperature increase (dT/dt) of 1 °C s$^{-1}$. At this juncture, the batteries are prone to ignition within moments, hence it is termed the thermal runaway temperature. T3 signifies the peak temperature reached during a thermal runaway event, serving as a key indicator for evaluating the intensity of the thermal runaway in batteries.

As shown in Supplementary Fig. 40, the values of T1 for the pouch cells utilizing the flame-retardant electrolyte with TTE and C3-2Cl diluents are respectively 103.8 °C and 105.7 °C, surpassing that of 82.3 °C in EC/DEC-based electrolyte. Furthermore, the pouch cell with EC/DEC-based electrolyte exhibited a severe thermal runaway at a T2 of 104.3 °C (Fig. 6e), immediately resulting in a high heat leakage with a high T3 of 592.5 °C and causing a massive gas leakage of 7.92 L (Supplementary Fig. 41a), which leads to severe burnout of the battery. Interestingly, the pouch cells with the flammable-retardant electrolytes displayed no serious thermal runaway with less gas production (Supplementary Fig. 41b and c), and the structure of pouch cells remained relatively integral after ARC test (Supplementary Fig. 42). It is found that the pouch cell with LiFSI-TEP/C3-2Cl exhibits an initial voltage drop at 112.3 °C and completely short-circuited at a high temperature of 293.2 °C (Supplementary Fig. 43a), these temperatures

are much higher than the values obtained in LiFSI-TEP/TTE (Supplementary Fig. 43b). Notably, the above results are inconsistent with those reported in previous papers that the use of flame retardant electrolytes based on phosphate solvents and TTE diluent will also cause thermal runaway[43,44], which may be directly related to the differences in battery design parameters and the contents of electrolyte. To fig out the reasons, we refer to the previous paper and increase the electrolyte injection amount from 1.5 μL mAh$^{-1}$ to 3 μL mAh$^{-1}$ (close to 3 g Ah$^{-1}$ used in previous papers)[44,45]. As shown in Supplementary Fig. 41d, e, when the amount of electrolyte is high, the thermal runaway of the battery still occurs even if the flame-retardant electrolyte is used, which is consistent with the previous reports. Compared to the LiFSI-TEP/TTE electrolyte, the pouch cell with LiFSI-TEP/C3-2Cl electrolyte shows higher T1 temperature and lower gas production. Meanwhile, as shown in Fig. 6e, a T2 temperature of 202.1 °C was detected in the pouch cell with 3 μL mAh$^{-1}$ LiFSI-TEP/C3-2Cl, which is much higher than the T2 values obtained in 3 μL mAh$^{-1}$ LiFSI-TEP/TTE (154.3 °C) and even 1.5 μL mAh$^{-1}$ EC/DEC-based electrolyte (104.3 °C). Above results strongly suggest that the flame-retardant electrolyte with C3-2Cl diluent can significantly enhance the safety of LMBs. In addition, even under extreme safety test conditions of nail penetration short circuit (Supplementary Fig. 44) or overcharge abuse (Supplementary Fig. 45), the pouch cell with 1.5 μL mAh$^{-1}$ LiFSI-TEP/C3-2Cl can still guarantee low temperature rise with no apparent thermal runaway, demonstrating high safety for practical applications. According to the above results, an interesting finding is also reported in this work, i.e., the synergistic effect of flame retardant electrolyte and low electrolyte injection amount helps to suppress the thermal runaway problem of LMBs, which provides an effective reference for the design of high safety and practical LMBs.

## Discussion

In summary, we have revealed a strategy to develop wide-temperature and flame-retardant electrolytes for LMBs by utilizing electrochemically-inert and weakly coordinating dichloroalkane (C-2Cl) diluents. By comparing the effects of C-2Cl diluents with different carbon chain length on the solvation structure and redox behavior of the electrolyte, we have found that the short-chain diluents (carbon chain length ≤3) tend to form a stable cyclic chelate structure with Li$^+$, which shows good electrochemical inertness in the electrolyte. Among them, C3-2Cl diluent shows favorable redox stability and wide liquid temperature range (−99 to +120 °C). In the optimized electrolyte of LiFSI-TEP/C3-2Cl, the anions-derived inorganic-rich EEIs are both formed on the surfaces of Li metal negative electrode and Ni-rich positive electrode, while the C3-2Cl diluent is partly involved into the Li$^+$ solvation structure to improve the Li$^+$ transport/desolvation capability. Therefore, the high-voltage LMBs, pairing the flame-retardant electrolyte under practical extreme conditions, exhibit good cycling stability and high discharge capability over a wide temperature range from −60 to +60 °C. Meanwhile, this flame-retardant electrolyte also shows high safety performance at extreme safety tests. This work advances the understanding of the solvation behavior of chlorinated diluents with different molecular structures, and proposes a strategy for developing superior high-safety liquid electrolytes by utilizing electrochemically-inert and weakly coordinating diluents to simultaneously achieve good redox stability and good Li$^+$ transport/desolvation capability.

## Methods

### Materials

LiFSI (CAS: 171611-11-3, 99%+, DodoChem Co., Ltd.) was dried at 110 °C for 24 h in a vacuum chamber before use. TEP (CAS: 78-40-0, 99%+, Adamas), TTE (CAS: 16627-68-2, 98%+, Adamas), C1-2Cl (CAS: 75-09-2, 99%+, Adamas), C2-2Cl (CAS: 107-06-2, 99%+, Adamas), C3-2Cl (CAS: 142-28-9, 99%+, Adamas), C4-2Cl (CAS: 110-56-5, 98%+, Adamas), and

C5-2Cl (CAS: 628-76-2, 98%+, Adamas) were purchased from Titan Technology Exploration Platform (Shanghai Titan Technology Co., Ltd.) and dried with molecular sieves (3 Å) before use. The commercial carbonate-based electrolyte of 1 M LiPF$_6$-EC/DEC (1:1 by vol.) was provided by DodoChem Co., Ltd. Other electrolytes tested in this work were prepared by directly dissolving LiFSI in TEP solvent and different C-2Cl diluents in a glove box filled with argon gas (O$_2$ < 1 ppm, H$_2$O < 1 ppm). NCM523 and NCM83 positive electrode materials were purchased from Ronbay Lithium Battery Material Co., Ltd. Li composite Cu foils (50 μm Li) were obtained from China Energy Lithium Co., Ltd. They were precisely cut to the required size and then directly used for assembling the batteries.

### Preparation of NCM523 and NCM83 positive electrodes

NCM523 or NCM83 active material, Super P (SP, IMERYS) conductive agent, and polyvinylidene fluoride (PVDF, Arkema) binder were mixed following a weight ratio of 90:5:5 in N-methyl-2-pyrrolidone (NMP, anhydrous, Adamas), respectively. Then the slurry was magnetically stirred for 8 h. After that, the as-prepared slurry was coated on aluminum (Al) foil and vacuum-dried at 80 °C for 12 h. The electrodes for coin cells were punched into disk with a diameter of 12 mm and a mass loading of 12 mg cm$^{-2}$ (single-side coated) after vacuum dried. For pouch cells, the electrodes were punched into a rectangular piece with a size of 43mm × 63mm and a mass loading of 56mgcm$^{-2}$ (double-sided coated) after being vacuum-dried.

### Electrochemical tests

All electrochemical tests are based on three times measurements. All coin cells used in this work were assembled in CR2025 coin-type cells using polypropylene (PP) membrane (Celgard 2400) as the separator. LSV of Li||Al and CV of Li||Cu coin cells were measured by a VPM300 analyzer (Biologic, France) to identify the redox behavior of electrolyte, a scanning rate of 0.1 mV s$^{-1}$ was selected. The Li||Cu and Li||Li coin cells were used to measure the Li plating/stripping performance in different electrolytes. PEIS (Potentiostatic Electrochemical Impedance Spectroscopy) was carried out by a VPM300 analyzer in a frequency range from 7 MHz to 100 mHz, with an AC voltage amplitude of 10 mV. The measurement was conducted with 5 points per decade in logarithmic spacing. The applied quasi-stationary potential was set near the open-circuit voltage (OCV) of the tested coin cell. CTTA was performed using Li||Cu cells. During the titration step, a current of I(τ) = 10 μA cm$^{-2}$ was applied for 0.1 h, corresponding to a charge of q(τ) = 1 μAh cm$^{-2}$ was deposited on Cu foil. After each titration step, an OCV period began, during which the cell voltage (E) was monitored. As the titrated Li was gradually consumed by side reactions, the E gradually increased and the OCV period was terminated when E = 0.05 V. The whole process with the titration step and the OCV period was repeated until 400 h. The anodic corrosion of Al current collectors in various electrolytes was investigated using Li||Al cells with chronoamperometry (CA) tests at 4.5 V for 24 h. The NCM523 (2.0 mAh cm$^{-2}$)||Li (50 μm) coin cells (N/P ratio ~5, electrolyte amount ~25 μL mAh$^{-1}$) were tested at 25 °C under a voltage profile of 2.8 ~ 4.5 V. To assemble a Li||NCM83 pouch cell, NCM83 positive electrodes (5.6 mAh cm$^{-2}$) and Li composite Cu negative electrodes (50 μm Li) (N/P ratio ~1.8, electrolyte amount ~5 μL mAh$^{-1}$) were stacked layer-by-layer and encapsulated by Al-plastic film, then the electrolyte was injected into the cell and stood for 12 h. The pouch cells were tested under a voltage profile of 2.8 ~ 4.3 V at 25 °C, and a pre-activation process at 0.05 C charge-discharge rate was performed before cycling. For the low-temperature and high-temperature tests, the pouch cells were put into a temperature chamber (LS-GDH, Guangdong Lestest Equipment Co., Ltd.) and stored at the test temperature for at least 5 h before testing. All electrolytes for full cells were added 5 wt% fluoroethylene carbonate (FEC) to further strengthen the electrode

stability. All the above cells were tested using the Neware battery test system (CT-4008, Shenzhen Neware Technology Co., Ltd.). The Li$^+$ transference number ($t_{Li^+}$) was calculated by using the Bruce-Vincent method[8,46]:

$$t_{Li^+} = \frac{(I_s (\Delta V - I_0 R_0))}{(I_0 (\Delta V - I_s R_s))} \quad (1)$$

where $R_0, R_s$ are resistance values measured at initial-state and steady-state, respectively. $I_0, I_s$ are initial-state and steady-state current, respectively. $\Delta V$ of 10 mV is the applied constant potential during the polarization process. Above tests were performed by using Li/Li symmetric cells on VMP300 Analyzer at 25 °C.

### Characterizations

The Li solubility in different diluents was measured by the ICP-OES (PerkinElmer Avio 200), all the liquid solution was collected for acid digestion before testing. FTIR spectroscopy was performed to study the solvation ability of different C-2Cl diluents by using Thermo Scientific Nicolet 6700 spectrometer, the solution was directly tested using an attenuated total reflection (ATR) attachment. $^7$Li NMR was performed on a Bruker AVANCE NEO-600 instrument at 25 °C using CDCl$_3$ as a solvent, and the $^7$Li peaks were calibrated by using 1 M LiCl in D$_2$O as the internal reference at δ = 0 ppm. The viscosity was measured by an NDJ-9S viscometer (LICHEN-BX Co., Ltd.) at 25 °C. The contact angle (FCA2000A, AFES, China) was collected by placing a droplet of electrolyte onto the PP separator as follows: the PP separator was fixed to a plate, and the plate was placed on a custom-made goniometer; then, we used a microsyringe to dispense a droplet of electrolyte onto it. The side-view image of the droplet was captured and the contact angle was calculated along the three-phase boundary[47]. The ionic conductivity of the electrolyte under different temperatures was measured using the electrochemical impedance method, as well as using a METTLER TOLEDO S400 SevenExcellence instrument to calibrate the values. DSC was detected by a DSC analyzer (200 F3 Maia, NETZSCH, Germany) under a nitrogen atmosphere ranging from 20 °C to −80 °C with a freezing rate of 10 °C min$^{-1}$. Morphologies of the Li metal and NCM83 electrodes were observed using SEM (Regulus 8230) with a beam voltage of 5 kV after rinsing the samples in 1,2-dimethoxyethane (DME) and drying them in argon glovebox. TEM (JEOL JEM-F200) was performed to identify the CEI layer formed on the NCM83 particle surface. Cryo-TEM was carried out by a JEOL JEM-F200 microscope at a cryogenic temperature of −170 °C. Li metal was first plated onto carbon grids in coin cells at 0.2 mA cm$^{-2}$ for 30 min, then the grids were washed with DME and transferred to and sealed inside a Fischione 2550 frozen transfer rod in an Ar-filled glove box. The frozen transfer rod was quickly inserted into the TEM microscope, and then filled with liquid nitrogen to establish a stable temperature of −170 °C. XPS analysis was performed on an ESCALAB 250Xi spectrometer (VG, Altrincham, UK) with Al Kα radiation. The samples were washed by DME and dried in an argon glovebox before testing, then transferred into XPS analysis chamber by a vacuum transfer device to avoid direct contact with air and water. TOF-SIMS analysis (TOF. SIMS5-100) was performed by a 10 kV Bi$^+$ ions beam and 1 kV Cs$^+$ sputtering. The ARC was measured by the Extended Volume Adiabatic Acceleration Calorimeter (EV + ARC, THT), and the pouch cell was first charged to the cut-off voltage of 4.3 V at 0.1 C before the ARC test. An electric wire was fixed on the surface of the pouch cell to collect the temperature changes of the battery. The temperature gradient was set to 5 °C, and the waiting time was set to 60 min. Nail penetration test of the pouch cell was carried out by a battery extrusion needle testing machine (BE-6047-50T), a nail with a diameter of 5 mm was driven into the pouch cell at a rate of 25 ± 5 mm s$^{-1}$. An overcharging test was performed by charging the pouch cell to a high cut-off voltage (6.45 V) at 1 C, then kept at 6.45 V for 1 h. At the same

time, the temperature sensor probe was fixed on the center of the battery surface.

## Theoretical calculations

The ESP diagram and LUMO/HOMO energies of different molecules and complexes were obtained by DFT calculation, which was carried out by DMol3 module in Materials Studio software[48,49]. The $E_B$ of Li$^+$ with different solvents was calculated by the following formula: $E_B = E_{total} - E_1 - E_2$. $E_{total}$, $E_1$ and $E_2$ were the total energy of the Li$^+$-solvent, single Li$^+$ and single solvent, respectively[50]. The geometry optimizations of different molecules were carried out with generalized-gradient approximation (GGA)/Perdew–Burke–Ernzerhof (PBE) functional and double numeric polarization (DNP) basis set[32,51]. The convergence tolerance was set to be $1.0 \times 10^{-5}$ Ha, $2.0 \times 10^{-3}$ Ha Å$^{-1}$, and $5.0 \times 10^{-3}$ Å for energy, maximum force, and maximum displacement, respectively. Classical MD simulations were performed using Forcite package in Materials Studio[49]. The amorphous cells with $31.0 \times 31.0 \times 31.0$ Å$^3$, $31.6 \times 31.6 \times 31.6$ Å$^3$, $32.2 \times 32.2 \times 32.2$ Å$^3$, $33.1 \times 33.1 \times 33.1$ Å$^3$, $33.9 \times 33.9 \times 33.9$ Å$^3$ linear dimensions were adopted for MD simulations, corresponding to the LiFSI/TEP/C-2Cl with molar ratio of 40:60:80 for the electrolytes with C1-2Cl, C2-2Cl, C3-2Cl, C4-2Cl and C5-2Cl, respectively. The MD simulations were conducted using the COMPASSIII force field, with atom types and partial charges optimized based on prior validated studies[52–55]. The system was subsequently equilibrated in the isothermal-isobaric (NPT) ensemble for 200 ps (picoseconds) with a time step of 1.0 fs (femtoseconds). To maintain thermodynamic stability, a Nosé-Hoover thermostat and a Parrinello-Rahman barostat were employed[56,57], with the following parameters: Cell time constant: 1.0 ps; Compressibility: $4.5 \times 10^{-5}$ GPa$^{-1}$; Coupling mode: Isotropic. The temperature was held constant at 298.15 K, and the pressure was maintained at 1 bar throughout the simulation. Subsequently, production runs were performed in an NVT ensemble, and the temperature was controlled using a Nosé-Hoover thermostat with a target temperature of 298.15 K. A time step of 1.0 fs and total simulation time of 2 ns (nanosecond) were chosen.

## Data availability

The authors declare that the data supporting the findings of this study are available within the paper and its Supplementary Information files. Should any raw data files be needed in another format they are available from the corresponding author upon request. Source data are provided with this paper.

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

## Acknowledgements

This work was financially supported by the National Key R&D Program of China (Grant No. 2022YFE0207300), the National Natural Science Foundation of China (Grant Nos. 22179142 and 22075314), and Jiangsu Funding Program for Excellent Postdoctoral Talent (Grant Nos. 2024ZB051, 2023ZB836, and 2023ZB392).

## Author contributions

Z.W. and H.T. contributed equally to this work. Z.W., H.T., and J.X. conceived the project. Z.W., X.M., X.Y., and K.L. prepared the electrolytes and batteries. H.T., S.L., Y.G., J.X., L.L., K.W., F.Z., Z.Q. contributed to part of the preparation and characterization of the electrolyte and the electrodes. Z.W., H.T., and G.S. contributed to the theoretical calculations. J.Z., Q.W., J.X., L.C., H.L., and X.W. supervised the project. The manuscript was written through the contributions of all authors.

## Competing interests

The authors declare no competing interests.

## Additional information

[1]Tianmu Lake Institute of Advanced Energy Storage Technologies Co., Ltd., Liyang, P. R. China. [2]Beijing Advanced Innovation Center for Materials Genome Engineering Key Laboratory for Renewable Energy, Beijing Key Laboratory for New Energy Materials and Devices, Institute of Physics, Chinese Academy of Sciences, Beijing, P. R. China. [3]i-lab, Suzhou Institute of Nano-Tech and Nano-Bionics (SINANO), Chinese Academy of Sciences, Suzhou, P. R. China. [4]College of Material Science and Engineering, Hohai University, Changzhou, P. R. China. [5]Changzhou FIRS Battery Technology Co., Ltd., Liyang, P. R. China. [6]Department of Materials Science and Engineering, National University of Singapore, Singapore, Singapore. [7]Center of Materials Science and Optoelectronics Engineering, University of Chinese Academy of Sciences, Beijing, China. [8]These authors contributed equally: Zhicheng Wang, Haifeng Tu. [9]These authors jointly supervised this work: Jingjing Xu, Hong Li, Xiaodong Wu. ✉e-mail: jjxu2011@sinano.ac.cn; hli@iphy.ac.cn; xdwu2011@sinano.ac.cn

