## [Transparent Peer Review file · Nature Communications]

Flame-retardant electrolytes with electrochemically-inert and weakly coordinating dichloroalkane diluents for practical lithium metal batteries

Corresponding Author: Professor Jingjing Xu

Version 0:

Reviewer comments:

Reviewer #1

(Remarks to the Author)

This manuscript presents a method for modulating locally concentrated electrolytes using chlorinated alkanes as diluents. The authors found that C3-2Cl exhibited the best performance, which is interesting. However, I do not think the manuscript is up to the level of Nature Communications. My suggestions and comments are as follows:

1. The authors state that 'the benefits of adding diluents include non-flammability', and I believe it is highly likely that the non-flammability is attributed to the TEP rather than the diluents themselves. Therefore, I recommend that the authors conduct ignition testing on all pure diluents.
2. As far as I know, all chlorinated alkanes are highly carcinogenic and pose significant environmental hazards, which greatly increases the risks associated with their application. Therefore, for safety reasons, I recommend that the authors include pouch cell ARC testing and analyze the composition of any leakage gases.
3. The authors do not provide a clear mechanistic explanation for why the performance of Li-Cu cells initially improves and then declines as the number of carbon atoms in the diluents increases.
4. The cycle conditions and performance of Li-Cu cells are inferior to similar studies published in Nature Communications (Nat Commun 13, 5431 (2022)), lead to the lack of sophistication.
5. Compared to LiFSI-TEP, the LiFSI-TEP/C3-2Cl electrolyte shows a limited improvement in the cycling performance of the 4.5V NMC532-Li cell.
6. The critical characterization of the CEI composition for the cycled 4.5V NMC532 cathode is missed.
7. How is the cycling performance of the full cell at -60°C and 60°C?

Reviewer #2

(Remarks to the Author)

In this manuscript by Wang & Tu et al., LiFSI-TEP electrolyte with dichloroalkane (C-2Cl) diluents are investigated and their performance are compared with their non-diluted counterpart (and also a conventional carbonate based electrolyte). After carefully reading it, this reviewer thinks that the manuscript is not suitable for publication in Nature Communications. However, the reviewer acknowledges the considerable efforts made by the authors in this study and believes that the work deserves publication in a more specialized journal with a more particular focus on electrochemistry and batteries. Some critical comments will be shared with authors -which will hopefully help to improve their manuscript before submission to another journal.

It is clear that diluents with long chain length result in fast cell degradation, but this reviewer has still concerns about the proposed high (electro)chemical inertness of the electrolyte with short chain length. As authors also pointed out, viscosity and wettability of electrolytes also change considerably, and these two parameters are expected to have a significant impact on electrochemical performance of cells with lithium metal electrodes (e.g., due to morphological effects of lithium plating and stripping). As a result, the electrochemical testing aiming to investigate electrochemical stability towards lithium metal is not convincing enough (e.g., the results of relatively fast CV/LSV tests, or relatively fast cycling in Li metal based cells).

Therefore, the authors are advised to demonstrate the inertness of this electrolyte with additional tests such as full cell testing without Li metal (e.g., with graphite CE instead of Li metal in NCM-based cells), slow-rate cycling with preferentially long waiting times between charging and discharging (i.e., waiting in charged state), CTTA tests, and slow-rate LSV tests, etc.

Reviewer #3

(Remarks to the Author)

The authors present an investigation of new electrolyte solutions for lithium metal batteries, using a plethora of experimental and simulation techniques.

The novelty is the addition of a dichloroalkane to improve the stability of the electrolyte, the cyclability of cells, and to obtain a flame-retardant electrolyte solution.

However, the paper contains a few flaws, especially regarding the simulations, and the manuscript requires major revisions before it might be acceptable. Comments are as follows:

- 1) In the title, I think it should read "electrochemically-inert".
- 2) Page 12, line 356: Isn't the value of 1.91 mS/cm a very low value compared with established electrolyte solutions used in lithium ion batteries?
- 3) Page 13, line 368: The authors provide four significant figures for contact angles. From my experience, an accuracy up to one decimal place can be achieved at best. With the method used by the authors, I rather suspect it would be just to round the value to the nearest integer.
- 4) Directly related to the previous comment: The description of the measurement method for the contact angle on page 19, line 540, is insufficient. Please provide more detail.
- 5) The authors should provide CAS numbers for all used chemicals, e.g. in the Section "Materials" starting in line 499 on page 17.
- 6) Page 19, line 546: °C is missing after -80.
- 7) The molecular dynamics (MD) simulation times of 5 ps equilibration and 50 ps production are way too short! The authors must repeat these simulations with total times of at least a few ns. The radial distributions shown in Fig. 2b indicate insufficient sampling and the choice of a too wide bin size.
- 8) Page 20, line 580: Which barostat was used for the NPT simulations? Please also provide the control parameter(s) of the barostat.
- 9) The description of the used force field seems to be incomplete. The authors claim to have used COMPASS II (page 20, line 579). However, for instance, neither the COMPASS I nor COMPASS II paper contains information on parameters for the lithium cation. In general, the COMPASS force field has been developed for molecular liquids and crystals, but not for electrolytes.

Reviewer #4

(Remarks to the Author)

This manuscript presents a systematic study using dichloroalkane (C-2Cl) diluents in flame-retardant electrolytes for lithium metal batteries. The authors provide systematic investigations from molecular design to practical applications, combining theoretical calculations with comprehensive experimental characterizations. While the work shows promise in achieving wide-temperature operation and improved safety, several critical issues need to be addressed before publication. The major concerns include incomplete safety evaluation, unclear mechanistic explanations, and insufficient comparative studies with state-of-the-art systems.

1. Chloroalkanes have been previously used as diluents in LHCEs for Li metal batteries (Zhang et al. Nat. Commun. 2022, 13, 5431). The authors need to add more comprehensive discussion about the novelty of this work compared to the previous studies.
2. The stability of aluminum current collectors in chloride-containing electrolytes is a critical concern that has been overlooked in this work. Potentiodynamic polarization measurements and post-mortem analysis of Al current collectors are needed.
3. The superior performance of C3-2Cl compared to C1-2Cl and C2-2Cl contradicts the proposed mechanism. If shorter chain lengths form more stable cyclic chelating structures with Li⁺, C1-2Cl and C2-2Cl should theoretically show better performance. The authors need to provide a quantitative analysis of chelating structure stability and explain this apparent discrepancy in their mechanistic framework.
4. The manuscript presents conflicting arguments regarding coordination strength. On one hand, it suggests that weak coordination between diluent and Li⁺ is beneficial; on the other hand, it promotes the advantages of cyclic chelating structures, which imply stronger coordination. The authors must clarify this fundamental contradiction and explain how these competing effects contribute to the observed performance improvements.
5. The connection between chelating stability and Li salt solubility needs to be established clearly. The authors should present solubility data for LiFSI in different chain length diluents and explain how the chelating structures affect salt dissociation and ionic conductivity. This relationship is crucial for understanding the electrolyte's practical performance limitations.
6. A direct comparison with TTE-based electrolytes under identical conditions is necessary to validate the claimed advantages of this system. The authors should conduct comprehensive performance comparisons including Li CE, cycling stability, and temperature performance. This will help position their work within the context of current state-of-the-art electrolytes.
7. The current safety assessment is inadequate for practical applications. The authors need to conduct comprehensive safety tests including nail penetration, overcharge abuse studies, thermal stability analysis. These tests are essential for

demonstrating the practical viability of chlorine-containing electrolytes.

8. The stability of C-Cl bonds under high voltage conditions requires thorough investigation. In-situ gas analysis during cycling and quantification of potential chlorine evolution are necessary. The authors should examine the decomposition products after long-term cycling for C-Cl bond stability.

Version 1:

Reviewer comments:

Reviewer #1

(Remarks to the Author)

accept

Reviewer #2

(Remarks to the Author)

The responses by Wang & Tu et al. to my critical comments/suggestions are mostly satisfactory. However, there are still some points remaining for the authors should consider for revision.

1) It is interesting that the proposed electrolyte is able to form a good performing SEI for lithium metal, but not able to form a SEI providing protection against graphite exfoliation. This is an important observation for the readers to be aware of, and the authors should clearly mention these observations in the manuscript (they can share the results in SI or not, as they prefer).

2) CV/LSV/CTTA tests. Fig3a. In this test, the authors also decided to scan below 0 V vs. Li+/Li. As expected Li plating started in this region, resulting in high current, thereby masking in the graph the most relevant section above 0 V where side reactions occur at much lower currents. The most interesting region in such a CV/LSV experiment is the potential region above 0 V, where information about the stability of electrolyte could be achieved. The authors don't need to repeat the experiments, but they should modify this graph with a new axis range for both x and y axis. They can give the data only for the first cycle where X-axis should only show 0 - 1 V region, and the y-axis should be scaled accordingly. They can still use the existing graph as an additional graph in MS or SI. Figure S10-a. Please double check the units of charge as it seems to be wrong.

Reviewer #3

(Remarks to the Author)

The authors have improved the paper in many aspects, but still, a few major issues remain.

Numbers in the following refer to those of my previous review.

3) Simply providing a snapshot of a commercial software tool for contact angle determination does not prove the claimed accuracy of the contact angle measurements. The authors should rather put forward strong arguments why they still report four significant figures or round the results to the nearest integer.

8) The name of the barostat is mentioned, but its control parameters are still missing.

9) The authors now claim to have used COMPASSIII, whereas they reported to have used COMPASS II in the previous version of the paper. Please clarify. Also, please provide a reference where the actual values for the model parameters, e.g. those of Li+, have been reported. The provided screenshot is not helpful.

Reviewer #4

(Remarks to the Author)

The authors have made efforts to address the previous concerns; however, significant contradictions and inconsistencies remain unresolved in both the revised manuscript and the provided rebuttal. These issues fundamentally undermine the validity of the conclusions and render the manuscript unsuitable for publication. Major concerns are outlined below:

1. Contradiction Between Experimental Data and MD Simulation Results

The manuscript claims that chlorinated diluents participate in the inner solvation sheath of Li⁺ ions (~2 Å) based on MD simulations. However, this contradicts well-established literature (Zhang et al. Nat. Commun. 2022, 13, 5431) demonstrating that chlorinated diluents exhibit extremely weak coordination with Li⁺ (typical distances >4 Å). The authors' own experimental data further corroborate this inconsistency: the solubility of Li salts in chlorinated diluents is less than 6 mM (2–3 orders of magnitude lower than in the primary phosphate-based solvents). This suggests that chlorinated diluents are thermodynamically unfavorable for inner-shell coordination. In addition, the polar TEP molecule (triethyl phosphate), which has stronger coordination capability than ethers, would even further limit the coordination of chlorinated diluents with Li⁺.

2. Negligible Impact of Diluent Coordination on Electrochemical Performance

Even if a minor fraction of chlorinated diluents coordinates with Li⁺ (which remains unproven), the proportion would be vanishingly small due to their ultra-low solubility and weak coordination strength. Such a negligible contribution cannot plausibly account for the observed electrochemical improvements (e.g., cycling stability or Li-metal compatibility). Instead, the intrinsic reactivity differences between uncoordinated diluent molecules (e.g., stability against Li metal or electrolyte decomposition pathways) are far more likely to dominate performance outcomes. Regrettably, the authors entirely omit discussions or experiments addressing these critical factors, rendering their mechanistic interpretation incomplete and potentially misleading.

3. Discrepancies in ARC Thermal Runaway Results

The manuscript reports that cells using chlorinated diluents or TTE exhibit suppressed thermal runaway (no significant dT/dt increase), which starkly conflicts with recent studies on phosphate ester-based localized high-concentration electrolytes (LHCEs). Multiple independent works (including systems with TEP solvents or TTE diluents) consistently demonstrate that thermal runaway persists in such systems. The authors' anomalous results raise serious concerns about the validity of their testing methodology. These unresolved discrepancies cast doubt on the reliability of the ARC data and the broader conclusions drawn from them.

Ref 1. Chen et al. *Advanced Materials*. 2024, 36, 2312302.

Ref 2. Ouyang et al. *Nat. Commun.* 2020, 11, 5100.

Ref 3. Li et al. *Journal of Energy Chemistry*, 2024, 97, 156.

Version 2:

Reviewer comments:

Reviewer #2

(Remarks to the Author)

I appreciate the revisions made by the authors. The responses are partially successful in resolving some concerns I have had, however, new results the authors provided also came with some new confusion/concerns. "I vs. E" data from CV tests for 0-2V region, which is now visible in the new graph, shows that LiFSI-TEP, C4-2Cl and C5-2Cl electrolytes have the highest stability against reduction. On the other hand, other electrolytes (including C3-2Cl) seem to be more reactive, with reactivity increasing from C3 to C2 and C1. This is not in agreement with the trends of other electrochemical tests. Both from the CV and CTTA data, it seems like that the LiFSI-TEP electrolyte is the most stable electrolyte from reduction stability perspective. The authors state in their conclusion that "C3-2Cl diluent shows outstanding redox stability...". Even though it is clear that this electrolyte results in good cycling performance in Li-Li cells and NCM-Li cells, it is not clear if a better electrochemical stability of this electrolyte is responsible for the better performance of symmetrical cells and half cells with this electrolyte. The authors also stated that the proposed electrolyte is not able to passivate a graphite electrode. Therefore, I would recommend the authors to provide more support with more comprehensive electrochemical analysis on their claims about the reduction stability of their electrolytes. This would require: i) more comprehensive CE tests in full cells with limited Li inventory (e.g., NCM-Cu anode free cells, NCM-Si full cells, etc.) performed at RT and elevated temperature at low C-rates, ii) CTTA tests performed directly (i.e., without high current/charge initial step as authors applied), iii) potentiostatic holding tests in anode-free cells to observe the charge accumulation at a potential near the lithium metal potential, e.g., holding at 0.1 V for 50 hours, etc.

Reviewer #3

(Remarks to the Author)

The authors have addressed my concerns adequately. Hence, depending on the opinion of the other reviewers and the editor, the paper may now be accepted.

Reviewer #4

(Remarks to the Author)

The authors have supplemented additional data and discussion for their arguments. The work could be accepted for publication.

Version 3:

Reviewer comments:

Reviewer #2

(Remarks to the Author)

The new experiments, e.g., potentiostatic holding tests at 0.1V as well as CE tests in Li inventory-limited full cells, do provide additional support and help to address the reviewer's concerns. At this stage, the reviewer thinks that the manuscript is now suitable for publication.

Point-by-point responses to reviewers' comments

Reviewer #1: *This manuscript presents a method for modulating locally concentrated electrolytes using chlorinated alkanes as diluents. The authors found that C3-2Cl exhibited the best performance, which is interesting. However, I do not think the manuscript is up to the level of Nature Communications. My suggestions and comments are as follows:*

Response: We deeply appreciate the reviewer for your professional suggestions and comments. We have incorporated additional experiments and analyses based on the reviewer's suggestions. Our point-by-point replies to the comments are detailed as follows.

Comment 1) *The authors state that 'the benefits of adding diluents include non-flammability', and I believe it is highly likely that the non-flammability is attributed to the TEP rather than the diluents themselves. Therefore, I recommend that the authors conduct ignition testing on all pure diluents.*

Response: We thank the reviewer for the valuable suggestion. We have supplemented the combustion tests of pure C-2Cl diluents with different chain-length. The results have been discussed in our revised manuscript. Besides, the flammability of pure diluents with different chain-length was also measured in Supplementary Fig. 19, showing that the flammability of the diluents increases with the increase of C/Cl ratio. As expected, when the carbon chain-length ≤ 3 , the diluents can exhibit rapid self-extinguishing characteristics, verifying the flame retardancy of short-chain C-2Cl diluents.

Supplementary Fig. 19 | Combustion tests of C-2Cl diluents with different chain-length. **a**, C1-2Cl, **b**, C2-2Cl, **c**, C3-2Cl, **d**, C4-2Cl, **e**, C5-2Cl.

Comment 2) *As far as I know, all chlorinated alkanes are highly carcinogenic and pose significant environmental hazards, which greatly increases the risks associated with their application. Therefore, for safety reasons, I recommend that the authors include pouch cell ARC testing and analyze the composition of any leakage gases.*

Response: We thank the reviewer for your professional suggestion. According to your comment, we have supplemented the thermal runaway performance and gas production of pouch cells with different electrolytes by ARC testing. Detailed discussions are shown in our revised manuscript.

In order to further investigate the safety performance of the flame-retardant electrolytes, 2 Ah-level NCM83||Li pouch cells (N/P ratio ~ 1.8 , electrolyte amount $\sim 1.5 \mu\text{L mAh}^{-1}$) with practical gravimetric energy density of 418.8 Wh kg^{-1} were assembled (Supplementary Fig. 35). Accelerated rate calorimetry (ARC) was performed to comprehensively evaluate the safety of pouch cells utilizing different electrolytes. Three key temperatures are crucial for identifying the safety performance of batteries. The onset temperature of self-heating, denoted as T1, marks the beginning of internal chain reactions that lead to an autonomous and sustained increase in temperature. Once the temperature surpasses T1, it becomes unsafe to operate or store the batteries. T2 is identified as the threshold temperature for thermal runaway, defined by a rate of

temperature increase (dT/dt) of $1\text{ }^{\circ}\text{C s}^{-1}$. At this juncture, the batteries are prone to ignition within moments, hence it is termed the thermal runaway temperature. T3 signifies the peak temperature reached during a thermal runaway event, serving as a key indicator for evaluating the intensity of the thermal runaway in batteries. As shown in Supplementary Fig. 36, the values of T1 for the pouch cells utilizing the flame-retardant electrolyte with TTE and C3-2Cl diluents are respectively $99.6\text{ }^{\circ}\text{C}$ and $105.7\text{ }^{\circ}\text{C}$, surpassing that of $82.3\text{ }^{\circ}\text{C}$ in EC/DEC-based electrolyte. Furthermore, the pouch cell with EC/DEC-based electrolyte exhibited a severe thermal runaway at a T2 of $99.3\text{ }^{\circ}\text{C}$ (Fig. 6g), immediately resulting in a high heat leakage with a high T3 of $592.5\text{ }^{\circ}\text{C}$ and causing a massive gas leakage of 7.92 L (Supplementary Fig. 36a), which leads to severe burnout of the battery. Interestingly, the pouch cells with the flammable-retardant electrolytes displayed no serious thermal runaway with less gas production (Supplementary Fig. 36), and the structure of pouch cells remained relatively integral after ARC test (Supplementary Fig. 37). It is found that the pouch cell with LiFSI-TEP/C3-2Cl exhibits an initial voltage drop at $112.3\text{ }^{\circ}\text{C}$ and completely short-circuited at a high temperature of $293.2\text{ }^{\circ}\text{C}$ (Fig. 6g), these temperatures are much higher than the values obtained in LiFSI-TEP/TTE (Supplementary Fig. 38). Above results strongly suggest that the flame-retardant electrolyte with C3-2Cl diluent can significantly enhance the safety of LMBs.

Fig. 6 | g, ARC tests of NCM83||Li pouch cells with different electrolytes.

Supplementary Fig. 36 | ARC results with temperature-time and gas production-time curves of NCM83||Li pouch cells in **a**, LiPF₆-EC/DEC, **b**, LiFSI-TEP/TTE, **c**, LiFSI-TEP/C3-2Cl.

Supplementary Fig. 37 | Optical images of pouch cells after ARC test in **a**, LiFSI-TEP/TTE and **b**, LiFSI-TEP/C3-2Cl.

Supplementary Fig. 38 | ARC results with dT/dt-temperature and voltage-temperature curves of NCM83||Li pouch cell in LiFSI-TEP/TTE.

Comment 3) *The authors do not provide a clear mechanistic explanation for why the performance of Li-Cu cells initially improves and then declines as the number of carbon atoms in the diluents increases.*

Response: We thank the reviewer for the precious question. We deeply apologize for our previous unclear descriptions about this section. We have added a more comprehensive discussion about the possible mechanism of the SEI chemistry on Li metal surface in electrolytes with different chain length C-2Cl diluents in our revised manuscript. Detailed discussions are shown below.

In conjunction with the above results and theoretical calculations discussed above, the possible mechanism of the SEI chemistry on Li metal surface in electrolytes with different chain length C-2Cl diluents is shown in Supplementary Fig. 18. It can be concluded that the long-chain diluent readily forms a high-valent 2Li^+ -diluent coordination complex with low LUMO energy, which induces the preferential decomposition of diluent to form an unstable SEI with high Cl-containing species on Li metal surface, thereby causes low Coulombic efficiency and failure of LMBs due to the persistent interfacial side reactions and electrolyte depletion. In contrast, the short-chain diluent benefits to form a low-valent 1Li^+ -diluent coordination complex with high LUMO energy, especially the C3-2Cl diluent readily forms a more stable hexatomic chelating structure with Li^+ , which possesses better reduction stability than pentatomic and tetratomic chelating structures formed by C2-2Cl and C1-2Cl diluents, respectively. Meanwhile, the anion-dominated Li^+ solvation structure formed in the electrolyte with C3-2Cl diluent can form a robust anion-derived inorganic-rich SEI layer on the Li metal electrode surface, effectively reducing the interfacial side reactions in LMBs and significantly enhancing their electrochemical performance.

Supplementary Fig. 18 | Mechanism diagram of the solvation structure and SEI chemistry in electrolytes with short-chain C-2Cl diluent or long-chain C-2Cl diluent.

Comment 4) *The cycle conditions and performance of Li-Cu cells are inferior to similar studies published in Nature Communications (Nat Commun 13, 5431 (2022)), lead to the lack of sophistication.*

Response: We deeply appreciate the reviewer to point out this deficiency. We are very regret that the performance of the electrolyte in this work is worse than that of the electrolyte in this reference (Nat. Commun. 13, 5431 (2022)) in terms of the Coulombic efficiency of Li||Cu cells. We believe that the main reason for this difference is that we chose the more reactive TEP solvent rather than the more Li metal-friendly DME solvent used in this reference. In our work, the main purpose is to develop a good electrolyte balancing high safety and high performance which is very important for lithium metal battery, hence we preferred TEP solvent with good flame retardant in order to ensure the intrinsic safety of the electrolyte. Another difference is that C3-2Cl diluent with wider liquid temperature range and higher electrochemical stability has been selected from the structural design of C-2Cl diluent in this work. To address your concerns, as shown in Figure R1 below, we have compared the performance of Li||Cu cells in LiFSI-DME with C2-2Cl or C3-2Cl diluents at the same conditions reported in this reference. As expected, it can be found that a high average Coulombic efficiency of ~99.3% was obtained both in these two electrolytes, indicating that the optimal C3-2Cl diluent also has a good application potential in other electrolyte systems to make improved electrochemical performance.

Figure R1 | Coulombic efficiency of Cu||Li cells in LiFSI-DME with C2-2Cl or C3-2Cl diluents at 0.5 mA cm^{-2} and 1 mAh cm^{-2} .

Comment 5) *Compared to LiFSI-TEP, the LiFSI-TEP/C3-2Cl electrolyte shows a limited improvement in the cycling performance of the 4.5V NMC532-Li cell.*

Response: We thank the reviewer for the precious question. As reported by other relevant literatures (Nat. Energy 3, 674–681 (2018); ACS Energy Lett. 4, 483–488 (2019); Joule 2, 1548–1558 (2018)), TEP-based high-concentration electrolyte benefits to form a stable anion-derived SEI layer on Li metal surface, thereby exhibits stable cycling performance in LMBs. Therefore, compare to high-concentration LiFSI-TEP electrolyte, the LiFSI-TEP/C3-2Cl electrolyte shows a limited improvement in the cycling performance in LMBs at a low charge-discharge rate of 0.5C. However, due to the high viscosity and low ionic conductivity of the LiFSI-TEP electrolyte, it is difficult to apply under the conditions of high-rate charging-discharging and high-loading electrodes. As shown in Fig. 6c and Fig. 6e in this work, it is difficult to use the LiFSI-TEP to exert sufficient capacity under these conditions, while the optimization of C3-2Cl diluent can significantly improve the discharge capacity of the batteries.

Comment 6) *The critical characterization of the CEI composition for the cycled 4.5V NMC532 cathode is missed.*

Response: We thank the reviewer for the professional suggestion. According to your comment, we have supplemented the TEM, XPS and TOF-SIMS characterizations for

the CEI layer formed on the cycled 4.5 V NCM523 cathode. The results have been discussed in our revised manuscript.

To compare the difference in CEI layer at higher cut-off voltage, 4.5 V NCM523 cathodes were also disassembled and characterized from coin cells after 50 cycles in TEP-based electrolytes with/without C3-2Cl diluent (Supplementary Fig. 32). Compare to the CEI layer formed by LiFSI-TEP, a thin (~5 nm) and uniform CEI layer was also observed on the surface of 4.5 V NCM523 cathode in LiFSI-TEP/C3-2Cl, which consists of negligible Cl-contained species and less C/P-contained species derived by C3-2Cl diluent and TEP solvent. These results indicate that C3-2Cl diluent is also electrochemically inert and can't be decomposed at high voltage.

Supplementary Fig. 32 | Characterizations of CEI layers formed on the 4.5 V NCM523 cathodes after 50 cycles. TEM of CEI layers formed by **a**, LiFSI-TEP/C3-2Cl and **b**, LiFSI-TEP. XPS spectra of CEI layers formed by **c**, **e**, **g**, LiFSI-TEP/C3-2Cl and **d**, **f**, **h**, LiFSI-TEP. TOF-SIMS of **i**, F^- , **j**, C_2HO^- and **k**, PO^- species in CEI layers formed by different electrolytes.

Comment 7) How is the cycling performance of the full cell at $-60^\circ C$ and $60^\circ C$?

Response: We thank the reviewer for the precious question. We have supplemented the cycling performance of the NCM83||Li pouch cell with optimized LiFSI-TEP/C3-2Cl

electrolyte at $-60\text{ }^{\circ}\text{C}$ and $60\text{ }^{\circ}\text{C}$. The results have been discussed in our revised manuscript.

Additionally, even charge/discharge processes were both operated at $60\text{ }^{\circ}\text{C}$ high temperature or $-60\text{ }^{\circ}\text{C}$ ultra-low temperature, NCM83||Li pouch cell with the electrolyte containing the C3-2Cl diluent still exhibited stable charge-discharge profiles during several cycles, manifesting the application potential of this electrolyte in wide temperature range (Supplementary Fig. 34).

Supplementary Fig. 34 | a, $60\text{ }^{\circ}\text{C}$ high-temperature and b, $-60\text{ }^{\circ}\text{C}$ low-temperature cycling performance of NCM83||Li pouch cell with LiFSI-TEP/C3-2Cl.

Reviewer #2: *In this manuscript by Wang & Tu et al., LiFSI-TEP electrolyte with dichloroalkane (C-2Cl) diluents are investigated and their performance are compared with their non-diluted counterpart (and also a conventional carbonate based electrolyte). After carefully reading it, this reviewer thinks that the manuscript is not suitable for publication in Nature Communications. However, the reviewer acknowledges the considerable efforts made by the authors in this study and believes that the work deserves publication in a more specialized journal with a more particular focus on electrochemistry and batteries. Some critical comments will be shared with authors -which will hopefully help to improve their manuscript before submission to another journal.*

Response: We sincerely appreciate the reviewer for your kind acknowledgement and valuable comments. Your professional suggestions and comments are very helpful for us to improve the quality of our manuscript. We have incorporated additional

experiments and analyses based on the reviewer's suggestions, and the detailed responses are as follows. Additionally, Nature Communications is a comprehensive journal to our knowledge, and it has published a lot of work on batteries and electrolytes similar to our work, so we believe that the quality of our paper will be greatly improved after revising according to your suggestions, and will be more suitable for Nature Communications readers.

Comment 1) *It is clear that diluents with long chain length result in fast cell degradation, but this reviewer has still concerns about the proposed high (electro)chemical inertness of the electrolyte with short chain length. As authors also pointed out, viscosity and wettability of electrolytes also change considerably, and these two parameters are expected to have a significant impact on electrochemical performance of cells with lithium metal electrodes (e.g., due to morphological effects of lithium plating and stripping). As a result, the electrochemical testing aiming to investigate electrochemical stability towards lithium metal is not convincing enough (e.g., the results of relatively fast CV/LSV tests, or relatively fast cycling in Li metal based cells). Therefore, the authors are advised to demonstrate the inertness of this electrolyte with additional tests such as full cell testing without Li metal (e.g., with graphite CE instead of Li metal in NCM-based cells), slow-rate cycling with preferentially long waiting times between charging and discharging (i.e., waiting in charged state), CTTA tests, and slow-rate LSV tests, etc.*

Response: We thank the reviewer for valuable suggestions. Firstly, to address your concerns about the impact of viscosity and wettability of electrolytes with different chain length diluents on electrochemical performance of LMBs, we have supplemented the viscosity tests of LiFSI-TEP with different chain length C-2Cl diluents. As shown in Supplementary Fig. 8, it is found that the similar viscosity is achieved in these electrolytes, indicating that the viscosity is not a crucial factor for the difference of electrochemical performance of LMBs in these systems.

Supplementary Fig. 8 | Ionic conductivity and viscosity of electrolytes with different diluents.

Secondly, we have assembled a NCM523||graphite (Gr) pouch cell with Gr anode instead of Li metal anode, using the LiFSI-TEP/C3-2Cl electrolyte with a charge-discharge of 0.05C at room temperature. Regretfully, this pouch cell showed a low discharge capacity at initial cycle (Figure R2a), indicating some undesired reactions are occurred in this system. Then, the cycled pouch cell was disassembled, showing obvious black reaction mottling and powder dropping phenomena on the surface of Gr electrode (Figure R2b). The XRD spectra demonstrate that the intensity of typical Gr diffraction peak (002) at 26.5° significantly decreases after first cycle (Figure R2c), suggesting the structure of Gr is destroyed by the co-intercalation or reaction of the electrolyte. To further verify above conjecture, morphologies of the pristine and cycled Gr electrodes were observed by SEM. It is found that a compact and flat Gr particle was observed in pristine electrode (Figure R2d), but it turns to an exfoliated layered structure after cycling (Figure R2e). The above results prove that this kind of electrolyte is not suitable for Li-ion battery system with Gr anode at this stage, and further research on its reaction mechanism in this system is needed in the follow-up work.

Figure R2 | Study on LiFSI-TEP/C3-2Cl electrolyte in NCM523||Gr pouch cell. **a**, Initial charge-discharge profile of pouch cell at 0.05C. **b**, Optical image of cycled Gr anodes and separators assembled from pouch cell. **c**, XRD spectra of pristine and cycled Gr electrodes. SEM images of the morphology of **d**, pristine Gr and **e**, cycled Gr.

Thirdly, according to the reviewer's comments, in order to further prove the inertness of the electrolytes, we have supplemented the slow-rate CV/LSV tests, CTTA tests, slow-rate cycling tests of 4.5 V NCM523||Li cells with preferentially long waiting times at charged state in our revised manuscript. As shown in Fig. 3a and Supplementary Fig. 11, the CV and LSV tests were both conducted using a slow scanning rate of 0.1 mV s^{-1} . The results are consistent with the previous tests, showing a high Li metal plating/stripping reversibility and a wide electrochemical window above 5 V in the LiFSI-TEP/C3-2Cl.

Fig. 3 | a, CV curves of Cu||Li cells with different electrolytes under a scanning rate of 0.1 mV s^{-1} .

Supplementary Fig. 11 | LSV curves of Al||Li cells with different electrolytes under a scanning rate of 0.1 mV s^{-1} .

Additionally, coulometric titration time analysis (CTTA) with Cu||Li cell was performed to quantify the side reactions between electrolyte and Li metal electrode ^[39]. As shown in Supplementary Fig. 10, significantly higher accumulated charges of Cu||Li cells are obtained in the electrolytes with long-chain C4-2CI and C5-2CI diluents, indicating severe side reactions between these electrolytes and Li metal. As expected, the electrolytes with short-chain diluents show less accumulated charge in Cu||Li cells, especially the electrolyte with C3-2CI diluent exhibits the lowest accumulated charge than other two short-chain diluents, verifying the lowest side reactions and consisting

with the best cycling performance in the electrolyte with C3-2Cl diluent.

Supplementary Fig. 10 | a, Accumulated charges of Cu||Li cells with different electrolytes. Voltage-time curves of Cu||Li cells with b, LiFSI-TEP, c, LiFSI-TEP/C1-2Cl, d, LiFSI-TEP/C2-2Cl, e, LiFSI-TEP/C3-2Cl, f, LiFSI-TEP/C4-2Cl, g, LiFSI-TEP/C5-2Cl.

Meanwhile, the slow-rate (0.05C) cycling with 1 h waiting time at 4.5 V charged state in NCM523||Li cell with LiFSI-TEP/C3-2Cl was also tested (Supplementary Fig. 26), which reveals almost no attenuation capacity within initial 20 cycles, indicating a high stability of the electrolyte in this system.

Supplementary Fig. 26 | Charge-discharge curves of NCM523||Li coin cell with LiFSI-TEP/C3-2Cl at initial 20 cycles with 0.05C rate and waiting 1 h in 4.5 V charged state.

Reviewer #3 *The authors present an investigation of new electrolyte solutions for lithium metal batteries, using a plethora of experimental and simulation techniques. The novelty is the addition of a dichloroalkane to improve the stability of the electrolyte, the cyclability of cells, and to obtain a flame-retardant electrolyte solution.*

However, the paper contains a few flaws, especially regarding the simulations, and the

manuscript requires major revisions before it might be acceptable. Comments are as follows:

Response: We sincerely appreciate the reviewer for your valuable comments. Your kind reminder and professional suggestions are very helpful for us to improve the quality of our manuscript. Our point-by-point replies to the comments are detailed as follows.

Comment 1) *In the title, I think it should read "electrochemically-inert".*

Response: We deeply appreciate the reviewer for reminding us this mistake. We carefully reviewed the full text and revised this description in both the title and the full text.

Comment 2) *Page 12, line 356: Isn't the value of 1.91 mS/cm a very low value compared with established electrolyte solutions used in lithium ion batteries?*

Response: We thank the reviewer for your precious question. To our best knowledge, a conductivity value of 1.91 mS cm⁻¹ is exactly relatively low compared to the typical electrolyte solutions used in lithium-ion batteries, but this value is higher than that of many reported advanced liquid electrolyte systems (e.g. ~1.29 mS cm⁻¹ of LiFSI-TEP/BTFE (Joule 2, 1548–1558 (2018)); ~1 mS cm⁻¹ of LiFSI-DEE (Nat. Energy 6, 303–313 (2021)); ~0.7 mS cm⁻¹ of LiFSI-DEM (ACS Appl. Mater. Interfaces 14, 44470–44478 (2022), etc.), indicating that it can provide sufficient ion transport capacity in practical applications.

Additionally, as we mentioned in our manuscript, **although the electrolyte with C3-2Cl diluent has a relative lower ionic conductivity (1.91 mS cm⁻¹) than that of the commercial carbonate electrolyte (7.53 mS cm⁻¹) at room temperature,** it shows a high ionic conductivity (0.60 mS cm⁻¹) at –20 °C, exceeding that of the commercial carbonate-based electrolyte (0.21 mS cm⁻¹). Even at –60 °C, the ionic conductivity of the electrolyte with C3-2Cl diluent was significantly higher than that of the commercial electrolyte.

Comment 3) *Page 13, line 368: The authors provide four significant figures for contact*

angles. From my experience, an accuracy up to one decimal place can be achieved at best. With the method used by the authors, I rather suspect it would be just to round the value to the nearest integer.

Response: We deeply thank the reviewer for the precious suggestion. We are very sorry that we miswrote the parameters of the test instrument previously, and we have corrected this mistake seriously in our revised manuscript. In this test, we captured the side-view image of the droplet and calculated the contact angle along the three-phase boundary, the value of the contact angle can be shown immediately in the AFES software, as shown in Figure R3 below.

Figure R3 | Screenshot of contact angle test software.

Comment 4) *Directly related to the previous comment: The description of the measurement method for the contact angle on page 19, line 540, is insufficient. Please provide more detail.*

Response: We thank the reviewer for the valuable suggestion. We have supplemented more detailed description of the measurement method for the contact angle in our revised manuscript, as shown below.

The contact angle (FCA2000A, AFES, China) was collected by placing a droplet of electrolyte onto the PP separator as follows: PP separator was fixed to a plate, and the plate was placed on a custom-made goniometer, then we used a microsyringe to

dispense a droplet of electrolyte on it. The side-view image of the droplet was captured and the contact angle was calculated along the three-phase boundary.^[43]

Comment 5) *The authors should provide CAS numbers for all used chemicals, e.g. in the Section "Materials" starting in line 499 on page 17.*

Response: We thank the reviewer for the valuable suggestion. We have supplemented the CAS numbers for all used chemicals in our revised manuscript.

LiFSI (CAS: 171611-11-3, DodoChem Co., Ltd.) was dried at 110 °C for 24 h in a vacuum chamber before use. TEP (CAS: 78-40-0), TTE (CAS: 16627-68-2), C1-2Cl (CAS: 75-09-2), C2-2Cl (CAS: 107-06-2), C3-2Cl (CAS: 142-28-9), C4-2Cl (CAS: 110-56-5), and C5-2Cl (CAS: 628-76-2) were purchased from Adamas and dried with molecular sieves before use.

Supplementary Table 1 | Physicochemical properties and market price of different diluents.

Solvent	CAS number	M. Wt	Melting point (T _m /°C)	Boiling point (T _b /°C)	Density (g cm ⁻³)	Market Price (Adamas, ¥ g ⁻¹)
Dichloromethane (C1-2Cl)	75-09-2	85	-97	40	1.3	0.36
1,2-Dichloroethane (C2-2Cl)	107-06-2	99	-35	84	1.2	0.29
1,3-Dichloropropane (C3-2Cl)	142-28-9	113	-99	120	1.1	0.46
1,4-Dichlorobutane (C4-2Cl)	110-56-5	127	-38	154	1.1	0.32
1,5-Dichloropentane (C5-2Cl)	628-76-2	141	-72	180	1.1	0.59
1,1,2,2-Tetrafluoroethyl-2,2,3,3-tetrafluoropropyl ether (TTE)	16627-68-2	232	-94	92	1.53	3.38

Comment 6) *Page 19, line 546: ° C is missing after -80.*

Response: We deeply appreciate the reviewer for your careful review and kind reminder. We have corrected this negligence in our revised manuscript.

Comment 7) *The molecular dynamics (MD) simulation times of 5 ps equilibration and 50 ps production are way too short! The authors must repeat these simulations with total times of at least a few ns. The radial distributions shown in Fig. 2b indicate insufficient sampling and the choice of a too wide bin size.*

Response: We sincerely appreciate the reviewer for your professional suggestion. We

have repeated the simulations and replaced the corresponding data in our revised manuscript, the regularities are found to be consistent with the previous discussion. Detailed calculation parameters and updated figures are shown below.

The amorphous cells with $31.0 \times 31.0 \times 31.0 \text{ \AA}^3$, $31.6 \times 31.6 \times 31.6 \text{ \AA}^3$, $32.2 \times 32.2 \times 32.2 \text{ \AA}^3$, $33.1 \times 33.1 \times 33.1 \text{ \AA}^3$, $33.9 \times 33.9 \times 33.9 \text{ \AA}^3$ linear dimensions were adopted for MD simulations, corresponding to the LiFSI/TEP/C-2Cl with molar ratio of 40:60:80 for the electrolytes with C1-2Cl, C2-2Cl, C3-2Cl, C4-2Cl and C5-2Cl, respectively. The solution structures were optimized with the COMPASSIII force field and then equilibrated in NPT using Nosé thermostat and a Parrinello-Rahman barostat to maintain a temperature of 298.15 K and a pressure of 1 bar.^[48,49] Afterward, production runs were performed in an NVT ensemble, and the temperature was controlled using a Nosé thermostat with a target temperature of 298 K. A time step of 1.0 fs (femtosecond) and total simulation time of 2 ns (nanosecond) were chosen.

Fig. 2 | Analysis of the electrolyte solvation structures with different C-2Cl diluents. a, ESP of different C-2Cl diluents and the coordination structures of Li⁺ with different C-2Cl diluents. **b**, RDF curves of Li-Cl coordination in electrolytes with different C-2Cl diluents. **c**, Percentages of different C-2Cl structures in different electrolytes obtained by MD simulations. The error bars represent the standard deviation from the mean and are defined as the average reading error from the data statistics. **d**, $\Delta\delta$ of C-Cl characteristic peaks in different electrolytes obtain by FTIR. **e**, $\Delta\delta$ of Li⁺ characteristic peaks in different electrolytes obtain by ⁷Li NMR.

Supplementary Fig. 2 | Snapshots of electrolyte solvation structures with different C-2Cl diluents obtained by MD simulations. **a**, LiFSI-TEP/C1-2Cl. **b**, LiFSI-TEP/C2-2Cl. **c**, LiFSI-TEP/C3-2Cl. **d**, LiFSI-TEP/C4-2Cl. **e**, LiFSI-TEP/C5-2Cl. Li⁺ and coordinated anions/solvents/diluents are shown in balls and sticks, free solvents/diluents are shown in lines. Color code: Li-purple, C-dark grey, H-white, O-red, N-blue, S-yellow, F-cyan, P-pink, Cl-green.

Supplementary Fig. 3 | RDF curves of Li⁺ coordination structures in different electrolytes obtained by MD simulations. **a**, LiFSI-TEP/C1-2Cl. **b**, LiFSI-TEP/C2-2Cl. **c**, LiFSI-TEP/C3-2Cl. **d**, LiFSI-TEP/C4-2Cl. **e**, LiFSI-TEP/C5-2Cl.

Comment 8) Page 20, line 580: Which barostat was used for the NPT simulations?

Please also provide the control parameter(s) of the barostat.

Response: We appreciate the reviewer for your precious question. We have supplemented the control parameter of barostat in the theoretical calculation section of our revised manuscript. Detailed information is shown below.

The solution structures were optimized with the COMPASSIII force field and then equilibrated in NPT using Nosé thermostat and a Parrinello-Rahman barostat to maintain a temperature of 298.15 K and a pressure of 1 bar.^[48,49]

Comment 9) *The description of the used force field seems to be incomplete. The authors claim to have used COMPASS II (page 20, line 579). However, for instance, neither the COMPASS I nor COMPASS II paper contains information on parameters for the lithium cation. In general, the COMPASS force field has been developed for molecular liquids and crystals, but not for electrolytes.*

Response: We deeply appreciate the reviewer for your valuable question. According to your comment, we have carefully checked the parameters contained in the COMPASSIII force field and found it contains the parameter information of lithium ion, as shown in Figure R4 below. Additionally, COMPASS force field has also been used in many references for electrolyte-related calculations, and our work is also based on these references (e.g. Nat. Commun. 15, 2033 (2024); Nat. Commun. 14, 8326 (2023); Adv. Energy Mater. 11, 2003905 (2021), etc.). In fact, we admit that the theoretical calculation itself will have a large error attribute, the calculated results need to be combined with the experimental results, and we believe that the calculated results can help us better understand the potential mechanism in some aspects.

Figure R4 | Screenshot of the parameter information in COMPASSIII force field.

Reviewer #4 *This manuscript presents a systematic study using dichloroalkane (C-2Cl) diluents in flame-retardant electrolytes for lithium metal batteries. The authors provide systematic investigations from molecular design to practical applications, combining theoretical calculations with comprehensive experimental characterizations. While the work shows promise in achieving wide-temperature operation and improved safety, several critical issues need to be addressed before publication. The major concerns include incomplete safety evaluation, unclear mechanistic explanations, and insufficient comparative studies with state-of-the-art systems.*

Response: We deeply appreciate the reviewer for your significant comments and suggestions. Your professional suggestions are very helpful for us to improve the quality of our manuscript. Our point-by-point responses to the comments are detailed as follows.

Comment 1) *Chloroalkanes have been previously used as diluents in LHCEs for Li metal batteries (Zhang et al. Nat. Commun. 2022, 13, 5431). The authors need to add more comprehensive discussion about the novelty of this work compared to the previous studies.*

Response: We appreciate the reviewer for your valuable suggestion. In our work, we

believe there are two major differences from this reference. First, the purpose of this work is to develop a liquid electrolyte system that can simultaneously take into account both high safety and wide temperature range applications. Therefore, TEP solvent with high flame retardancy is used instead of flammable DME solvent, which can ensure the safety of the electrolyte to a large extent. Second, the effects of a series of C-2Cl diluents with different chain lengths on the solvation structure, physicochemical properties and electrode interfacial chemistry were systematically compared and analyzed in this work. Consequently, C3-2Cl diluent with better performance was successfully selected, which was effectively employed into high-voltage LMBs to achieve high safety and impressive electrochemical performance over a wide temperature range of -60~60 °C. According to your suggestion, we also have supplemented more comprehensive discussion in contrast to this reference in the introduction section of our revised manuscript, detailed information is shown below.

Most recently, weakly polar symmetric dichloroalkane (C-2Cl) solvents such as 1,2-dichloroethane (C2-2Cl)^[34] have been reported as a diluent candidate in LHCE to enable high stable LMBs, which are expected to satisfy above two criteria as they can regulate electron cloud density via electronegative Cl atoms to weakly coordinate with Li⁺ and confer good flame retardancy by Cl element. However, due to the low boiling point (84 °C) and high melting point (-35 °C) of C2-2Cl diluent, it is difficult to meet the demands of wide temperature range electrolytes. As illustrated in Fig. 1b, a family of symmetric C-2Cl diluents with different carbon chain length (dichloromethane (C1-2Cl), C2-2Cl, 1,3-dichloropropane (C3-2Cl), 1,4-dichlorobutane (C4-2Cl), and 1,5-dichloropentane (C5-2Cl)) were identified as prime candidates, by comparing the melting and boiling points of the different C-2Cl diluents and the frequently utilized TTE diluent in the literature (Fig. 1c and Supplementary Table 1), it becomes evident that as the carbon chain length of the C-2Cl increases, the solvent's liquid temperature range (TR) widens. The C-2Cl solvents with a carbon chain length ≥ 3 exhibit promising potential for wide temperature range applications, with TR exceeding that of the TTE diluent. Moreover, these C-2Cl solvents also boast low density and low market price (Fig. 1d), demonstrating the advantages of large-scale application. Nevertheless,

the effects of C-2Cl diluents with different chain lengths on physicochemical properties, solvation structure and electrochemical properties of electrolytes remain unknown.

Comment 2) *The stability of aluminum current collectors in chloride-containing electrolytes is a critical concern that has been overlooked in this work. Potentiodynamic polarization measurements and post-mortem analysis of Al current collectors are needed.*

Response: We thank the reviewer for your professional suggestion. According to your comment, we have supplemented the experiments about the stability of Al current collectors in different electrolytes. Detailed discussions are shown in our revised manuscript.

To investigate the potential application of electrolytes in high-voltage LMBs, the stability of Al current collectors in different electrolytes was firstly investigated by using Al||Li cells with 4.5 V constant voltage charging for 24h (Supplementary Fig. 25). As reported in literature ^[19], the electrolyte with low LiFSI salt concentration (1 M LiFSI in TEP, namely LCE) shows severe corrosion to Al foil, which delivers high corrosion current under 4.5 V charging process and causes corrosion pitting on Al foil surface. In contrast, low corrosion currents and no obvious signs of corrosion were observed in high-concentration LiFSI-TEP (namely HCE) and LiFSI-TEP/C3-2Cl electrolytes, indicating a high stability of Al foils and good application potentials of these electrolytes in high-voltage batteries. XPS spectra on Al foil surface demonstrate that F-contained species derived from LiFSI salt corrosion were detected in LCE, while no obvious F-contained or Cl-contained species were formed in high-concentration LiFSI-TEP with/without C3-2Cl diluent, further verifying few corruptions between Al current collector and Cl-contained electrolytes.

Supplementary Fig. 25 | Stability of Al current collectors in different electrolytes. **a**, Current-time curves of Li||Al coin cells with different electrolytes under 4.5 V constant voltage charging for 24h. **b**, SEM images of the morphologies of Al foils after charging in different electrolytes. **c**, F 1s and Cl 2p XPS spectra of the surface of Al foils after charging in different electrolytes.

Comment 3) *The superior performance of C3-2Cl compared to C1-2Cl and C2-2Cl contradicts the proposed mechanism. If shorter chain lengths form more stable cyclic chelating structures with Li⁺, C1-2Cl and C2-2Cl should theoretically show better performance. The authors need to provide a quantitative analysis of chelating structure stability and explain this apparent discrepancy in their mechanistic framework.*

Response: We appreciate the reviewer for your valuable suggestion. We deeply apologize for our previous unclear descriptions about this section. According to our study, the hexatomic chelating structure of Li-C3-2Cl complex is more stable than the pentatomic chelating structure of Li-C2-2Cl complex and tetraatomic structure of Li-C1-2Cl complex, which exhibits higher electrochemical stability in electrolyte thereby enhances the performance of LMBs. The stability of these structures also has been quantitatively analyzed by DFT calculations and XPS spectra of the decomposition products on Li metal surface in our revised manuscript. Detailed discussions are shown below.

In addition, it is also found that the chelating structure of hexatomic Li-C3-2Cl

complex shows higher LUMO energy level than that of pentatomic Li-C2-2Cl complex or tetratomic Li-C1-2Cl complex, indicating a higher stability of the hexatomic chelating solvation structure and higher reduction stability of the electrolyte with C3-2Cl diluent.

Notably, the Li metal surface cycled in the electrolyte with C3-2Cl diluent shows the lowest content of Cl-containing components derived from diluent decomposition among these five C-2Cl diluents, as confirmed by the lowest atomic ratio of Cl obtained by XPS analysis (Supplementary Fig. 17), demonstrating a higher reduction stability of the electrolyte with C3-2Cl diluent.

In addition, we have added a more comprehensive discussion about the possible mechanism of the SEI chemistry on Li metal surface in electrolytes with different chain length C-Cl diluents in our revised manuscript. Detailed discussions are shown below.

In conjunction with the above results and theoretical calculations discussed above, the possible mechanism of the SEI chemistry on Li metal surface in electrolytes with different chain length C-2Cl diluents is shown in Supplementary Fig. 18. It can be concluded that the long-chain diluent readily forms a high-valent 2Li^+ -diluent coordination complex with low LUMO energy, which induces the preferential decomposition of diluent to form an unstable SEI with high Cl-containing species on Li metal surface, thereby causes low Coulombic efficiency and failure of LMBs due to the persistent interfacial side reactions and electrolyte depletion. In contrast, the short-chain diluent benefits to form a low-valent 1Li^+ -diluent coordination complex with high LUMO energy, especially the C3-2Cl diluent readily forms a more stable hexatomic chelating structure with Li^+ , which possesses better reduction stability than pentatomic and tetratomic chelating structures formed by C2-2Cl and C1-2Cl diluents, respectively. Meanwhile, the anion-dominated Li^+ solvation structure formed in the electrolyte with C3-2Cl diluent can form a robust anion-derived inorganic-rich SEI layer on the Li metal electrode surface, effectively reducing the interfacial side reactions in LMBs and significantly enhancing their electrochemical performance.

Supplementary Fig. 18 | Mechanism diagram of the solvation structure and SEI chemistry in electrolytes with short-chain C-2Cl diluent or long-chain C-2Cl diluent.

Comment 4) *The manuscript presents conflicting arguments regarding coordination strength. On one hand, it suggests that weak coordination between diluent and Li^+ is beneficial; on the other hand, it promotes the advantages of cyclic chelating structures, which imply stronger coordination. The authors must clarify this fundamental contradiction and explain how these competing effects contribute to the observed performance improvements.*

Response: We appreciate the reviewer for your precious question. We deeply apologize for your confusion about our previous unclear description. We have modified the description about the criteria for diluent selection in our revised manuscript.

In our opinion, the weakly coordinating ability and high electrochemical inertness set the secondary criteria for diluent selection, which is expected to enhance the Li^+ transport/de-solvation kinetics by partially coordinated with Li^+ , while easy to induce the preferential decomposition of anions due to the electrochemical inertness, thereby forming stable EEs on electrodes to improve the electrochemical performance of LMBs. The theoretical calculations and experimental results are consistent with above criteria in our work. On the one hand, more diluents participate in the Li^+ solvation structure not only effectively improves the ionic conductivity and Li^+ transference number of the electrolyte, but also reduces the de-solvation energy barrier of Li^+ , thereby improves the rate and low temperature performance of the battery. On the other hand, the cyclic chelating structures formed by diluents and Li^+ benefit to improve the electrochemical stability of diluents, which effectively suppresses the side reactions

between diluents and electrodes, thereby improves the cycling stability of the battery. Therefore, the stronger coordination ability and more stable cyclic chelating coordination structure between diluent and Li^+ can better fit the above criteria, benefiting for better electrochemical performance of electrolytes.

Comment 5) *The connection between chelating stability and Li salt solubility needs to be established clearly. The authors should present solubility data for LiFSI in different chain length diluents and explain how the chelating structures affect salt dissociation and ionic conductivity. This relationship is crucial for understanding the electrolyte's practical performance limitations.*

Response: We thank the reviewer for your professional suggestion. We have supplemented the Li salt solubility and ionic conductivity tests in our revised manuscript. Detailed discussions are shown below.

The influences of the coordination between diluents and Li^+ on the Li salt solubility and the ionic conductivity of electrolytes were investigated. Firstly, 0.1 M LiFSI salt was added in different pure C-2Cl diluents for 12 h stirring, then the Li solubility was obtained and quantified by inductively coupled plasma optical emission spectrometer (ICP-OES) (Supplementary Fig. 7). Even at this ultra-low Li concentration of 0.1 M, a large amount of undissolved Li salt can still be clearly observed at the bottom. ICP-OES results demonstrate that the Li salt solubility in pure diluent increases with the chain length of C-2Cl diluent extends, which is consistent with the above analysis of electrolyte solvation structures about more diluents participate in the Li^+ solvation structure with the increase in carbon chain length of the C-2Cl diluents. These results indicate that it is difficult to effectively dissociate sufficient Li salts in pure C-2Cl diluents, so C-2Cl diluents need to interact with TEP solvent to achieve electrolytes with effective ion transport properties. Through testing the viscosity and ionic conductivity of electrolytes containing different C-2Cl diluents, it is found that the similar viscosity is achieved in these electrolytes, but the ionic conductivity slightly increases first and then decreases with the increase of chain length (Supplementary Fig. 8). This may be affected by a combination of complex factors such as the solvation

energy of diluents on Li^+ and the volume of the Li^+ solvation sheath.^[38] The stronger solvation energy of long-chain diluents with Li^+ helps to improve the ionic conductivity, but the larger volume of Li^+ solvation sheath is not conducive to Li^+ transport and thus reduce ion conductivity.

Supplementary Fig. 7 | a, Optical image of 0.1 M LiFSI salt in different C-2CI diluents after 12 h stirring. b, Corresponding Li solubility in different C-2CI diluents obtained by ICP-OES.

Supplementary Fig. 8 | Ionic conductivity and viscosity of electrolytes with different diluents.

Comment 6) A direct comparison with TTE-based electrolytes under identical conditions is necessary to validate the claimed advantages of this system. The authors should conduct comprehensive performance comparisons including Li CE, cycling stability, and temperature performance. This will help position their work within the context of current state-of-the-art electrolytes.

Response: We thank the reviewer for your valuable suggestion. We have supplemented the electrochemical performance and physicochemical properties tests of the electrolyte with TTE diluent. Detailed discussions are shown in our revised manuscript.

Additionally, the state-of-the-art TTE diluent was also employed in LiFSI-TEP electrolyte (molar ratio of 1:1.5:3 in LiFSI-TEP/TTE) for comparison, which exhibits a slightly higher viscosity (7.1 mPa s) but lower contact angle (32.56°) due to the high surface tension of fluorine-contained TTE diluent. Furthermore, the Li^+ transference numbers of the different electrolytes were measured to identify the Li^+ transport property in bulk electrolyte (Supplementary Fig. 22). The addition of the C3-2Cl diluent increased the Li^+ transference number of the LiFSI-TEP electrolyte from 0.62 to 0.74, far exceeding the value (0.32) of commercial carbonate-based electrolyte and also higher than the value (0.68) of state-of-the-art LiFSI-TEP/TTE, indicating that the C3-2Cl diluent is beneficial for enhancing the Li^+ transport property in the electrolyte.

Supplementary Fig. 21 | Contact angle tests of different electrolytes. **a**, LiPF₆-EC/DEC, **b**, LiFSI-TEP, **c**, LiFSI-TEP/C3-2Cl, **d**, LiFSI-TEP/TTE.

Supplementary Fig. 22 | Li^+ transference number measurements of different electrolytes. **a**, $\text{LiPF}_6\text{-EC/DEC}$. **b**, LiFSI-TEP . **c**, LiFSI-TEP/TTE . **d**, LiFSI-TEP/C3-2Cl .

To study the potential application of the electrolyte in a wide temperature range, differential scanning calorimetry (DSC) was further used to test the freezing points of the three different electrolytes at low temperatures. During the cooling process, the commercial carbonate-based electrolyte exhibited a distinct exothermic peak at around $-40\text{ }^\circ\text{C}$ (Fig. 5c), corresponding to the electrolyte's freezing point. By contrast, no freezing point was observed in the other **three** flame-retardant electrolytes even when cooled to $-80\text{ }^\circ\text{C}$, indicating that these electrolytes can remain in a liquid state at ultra-low temperatures. Then, the ionic conductivity of these electrolytes over a wide temperature range was measured (Fig. 5d and Supplementary Fig. 23). **Although the electrolyte with C3-2Cl diluent has a relative lower ionic conductivity (1.91 mS cm^{-1}) than that of the commercial carbonate electrolyte (7.53 mS cm^{-1}) at room temperature,** it shows a high ionic conductivity (0.60 mS cm^{-1}) at $-20\text{ }^\circ\text{C}$, exceeding that of **the LiFSI-TEP/TTE (0.34 mS cm^{-1})**, **LiFSI-TEP (0.13 mS cm^{-1})** and the commercial carbonate-based electrolyte (0.21 mS cm^{-1}). Even at $-60\text{ }^\circ\text{C}$, the ionic conductivity of the electrolyte with C3-2Cl diluent was significantly higher than that of the other **three**

electrolyte systems. Furthermore, Li^+ de-solvation activation energy (E_a) was calculated based on the Arrhenius equation, which was measured by probing the charge transfer resistance (R_{ct}) of $\text{Li}||\text{Li}$ symmetric cells under different temperatures (Supplementary Fig. 24) [40,41]. By comparing the E_a of these three electrolytes (Fig. 5e), the electrolyte with C3-2Cl diluent also exhibited the lowest E_a value, indicating a faster Li^+ de-solvation capability in this electrolyte. The enhancement of Li^+ transport and de-solvation capability is mainly due to the weakly coordination ability between Li^+ and C3-2Cl diluent, which is different from TTE diluent with minimal coordination ability with Li^+ as reported by literatures [27,28].

Fig. 5 | Physicochemical properties of different electrolytes. **a**, Combustion tests of flammable cotton tapes soaked in different electrolytes. **b**, Viscosity of different electrolytes. Each bar stands for the mean of the maximum and minimum values of the 5 test results. **c**, DSC curves of different electrolytes at a freezing rate of $10\text{ }^{\circ}\text{C min}^{-1}$. **d**, Ionic conductivity of different electrolytes over a wide temperature range of -60 to $+60\text{ }^{\circ}\text{C}$. **e**, Li^+ de-solvation activation energy tests of different electrolytes.

Supplementary Fig. 23 | EIS analysis of stainless steel (SS)||SS symmetric cells under different temperatures with different electrolytes. **a**, LiPF₆-EC/DEC. **b**, LiFSI-TEP. **c**, LiFSI-TEP/TTE. **d**, LiFSI-TEP/C3-2Cl.

Supplementary Fig. 24 | EIS and equivalent circuit model of Li/Li symmetric cells at different temperatures with different electrolytes. **a**, LiPF₆-EC/DEC. **b**, LiFSI-TEP. **c**, LiFSI-TEP/TTE. **d**, LiFSI-TEP/C3-2Cl.

Consistent with the previous report ^[18,26], the electrolyte with TTE diluent also displayed superior cycling stability and high Coulombic efficiency in NCM523||Li cell. However, rate capability tests reveal that the electrolyte with C3-2Cl diluent effectively improved capacity utilization under high-rate charge/discharge conditions (Fig. 6c),

which is much better than the LiFSI-TEP with/without TTE diluent. Above results are mainly attributed to the weakly coordination between Li^+ and C3-2Cl diluent, which effectively enhances Li^+ transport/de-solvation characteristics.

Besides, the electrolyte with TTE diluent also exhibits stable cycling performance in NCM83||Li pouch cell, which is consistent with the previous study [18].

Compare to LiFSI-TEP electrolyte, although the low temperature performance of the NCM83||Li pouch cell has been improved after the addition of TTE diluent, the low temperature discharge capacity is still insufficient to meet the application requirements.

In summary, as shown in Fig. 6h, compared to the high-concentration LiFSI-TEP and state-of-the-art TTE diluent-optimized flame-retardant electrolyte systems reported for LMBs, the C3-2Cl diluent-optimized electrolyte in this work offers more advantages, such as lower cost, lower density, higher ionic conductivity, and a wider temperature application range. Additionally, this type of electrolyte also possesses a wide electrochemical window and high safety, effectively enhancing the rate capability and cycling performance of LMBs. Therefore, this flame-retardant electrolyte is expected to drive the large-scale application of high-energy-density, wide-temperature, long-life, and high-safety LMBs in the future.

Fig. 6 | Electrochemical performance of high-voltage LMBs with different electrolytes. a, Structural representation of NCM523||Li coin cell used in this work. **b,** Cycling performance of NCM523||Li coin cells with different electrolytes under test conditions of 2.8~4.5 V, 0.5C charge/discharge and room temperature. **c,** Rate capability of NCM523||Li coin cells with different electrolytes. **d,** Structural representation of NCM83||Li pouch cell used in this work. **e,** Cycling performance of NCM83||Li pouch cells with different electrolytes under test conditions of 2.8~4.3 V, 0.1C charge/0.2C discharge and room temperature. Inset is an optical image of the pouch cell. **f,** Discharge capacities of NCM83||Li pouch cells with different electrolytes under different temperatures. **g,** ARC tests of NCM83||Li pouch cells with different electrolytes. **h,** Comparison of the electrolytes obtained in this work for LMBs.

Comment 7) *The current safety assessment is inadequate for practical applications. The authors need to conduct comprehensive safety tests including nail penetration, overcharge abuse studies, thermal stability analysis. These tests are essential for demonstrating the practical viability of chlorine-containing electrolytes.*

Response: We appreciate the reviewer for your professional suggestion. According to your comment, we have supplemented the relevant safety tests to evaluate the safety performance of Cl-containing electrolyte in practical pouch cells. Detailed discussions are shown in our revised manuscript.

In order to further investigate the safety performance of the flame-retardant electrolytes, 2 Ah-level NCM83||Li pouch cells (N/P ratio ~1.8, electrolyte amount ~1.5 $\mu\text{L mAh}^{-1}$) with practical gravimetric energy density of 418.8 Wh kg^{-1} were assembled (Supplementary Fig. 35). Accelerated rate calorimetry (ARC) was performed to comprehensively evaluate the safety of pouch cells utilizing different electrolytes. Three key temperatures are crucial for identifying the safety performance of batteries. The onset temperature of self-heating, denoted as T1, marks the beginning of internal chain reactions that lead to an autonomous and sustained increase in temperature. Once the temperature surpasses T1, it becomes unsafe to operate or store the batteries. T2 is identified as the threshold temperature for thermal runaway, defined by a rate of temperature increase (dT/dt) of 1 $^{\circ}\text{C s}^{-1}$. At this juncture, the batteries are prone to ignition within moments, hence it is termed the thermal runaway temperature. T3 signifies the peak temperature reached during a thermal runaway event, serving as a key indicator for evaluating the intensity of the thermal runaway in batteries. As shown

in Supplementary Fig. 36, the values of T1 for the pouch cells utilizing the flame-retardant electrolyte with TTE and C3-2Cl diluents are respectively 99.6 °C and 105.7 °C, surpassing that of 82.3 °C in EC/DEC-based electrolyte. Furthermore, the pouch cell with EC/DEC-based electrolyte exhibited a severe thermal runaway at a T2 of 99.3 °C (Fig. 6g), immediately resulting in a high heat leakage with a high T3 of 592.5 °C and causing a massive gas leakage of 7.92 L (Supplementary Fig. 36a), which leads to severe burnout of the battery. Interestingly, the pouch cells with the flammable-retardant electrolytes displayed no serious thermal runaway with less gas production (Supplementary Fig. 36), and the structure of pouch cells remained relatively integral after ARC test (Supplementary Fig. 37). It is found that the pouch cell with LiFSI-TEP/C3-2Cl exhibits an initial voltage drop at 112.3 °C and completely short-circuited at a high temperature of 293.2 °C (Fig. 6g), these temperatures are much higher than the values obtained in LiFSI-TEP/TTE (Supplementary Fig. 38). Above results strongly suggest that the flame-retardant electrolyte with C3-2Cl diluent can significantly enhance the safety of LMBs.

In addition, even under extreme safety test conditions of nail penetration short circuit (Supplementary Fig. 39) or overcharge abuse (Supplementary Fig. 40), the pouch cell with the flame-retardant electrolyte containing C3-2Cl diluent can still guarantee low temperature rise with no apparent thermal runaway, demonstrating a high safety for practical applications.

Supplementary Fig. 35 | Optical image, overall weight and 0.05C charge-discharge profiles of 2 Ah-level pouch cell with LiFSI-TEP/C3-2Cl.

Fig. 6 | g, ARC tests of NCM83||Li pouch cells with different electrolytes.

Supplementary Fig. 36 | ARC results with temperature-time and gas production-time curves of NCM83||Li pouch cells in a, LiPF₆-EC/DEC, b, LiFSI-TEP/TTE, c, LiFSI-TEP/C3-2Cl.

Supplementary Fig. 37 | Optical images of pouch cells after ARC test in a, LiFSI-TEP/TTE and b, LiFSI-TEP/C3-2Cl.

Supplementary Fig. 38 | ARC results with dT/dt -temperature and voltage-temperature curves of NCM83||Li pouch cell in LiFSI-TEP/TTE.

Supplementary Fig. 39 | Nail penetration short circuit test and post-test optical image of 2 Ah-level NCM83||Li pouch cell with LiFSI-TEP/C3-2Cl.

Supplementary Fig. 40 Overcharge abuse test and post-test optical image of 2 Ah-level NCM83||Li pouch cell with LiFSI-TEP/C3-2Cl.

Comment 8) *The stability of C-Cl bonds under high voltage conditions requires thorough investigation. In-situ gas analysis during cycling and quantification of potential chlorine evolution are necessary. The authors should examine the decomposition products after long-term cycling for C-Cl bond stability.*

Response: We appreciate the reviewer for your professional suggestion. We have observed the gas production of the NCM83||Li pouch cell with LiFSI-TEP/C3-2Cl after 20 cycles, but did not see any significant gas production in the bag, as shown in Figure R5. Therefore, we further used ARC to in-situ detect the gas production of such cells under thermal runaway conditions. As shown in Supplementary Fig. 36, **the pouch cell with EC/DEC-based electrolyte exhibited a severe thermal runaway at a T2 of 99.3 °C (Fig. 6g), immediately resulting in a high heat leakage with a high T3 of 592.5 °C and causing a massive gas leakage of 7.92 L (Supplementary Fig. 36a).** Interestingly, **the pouch cells with the flammable-retardant electrolytes displayed no serious thermal runaway with less gas production (Supplementary Fig. 36).** Above results suggest that this flame-retardant electrolyte has a lower risk of gas production, demonstrating higher safety. It also should be noted that we are very sorry that due to our limited testing conditions, we cannot quantitatively analyze the chlorine content in the gas leakage. We will continue to improve relevant equipment for analysis in the future work.

Figure R5 | Front and side optical images of NCM83||Li pouch cell with LiFSI-TEP/C3-2Cl after 20 cycles.

Supplementary Fig. 36 | ARC results with temperature-time and gas production-time curves of NCM83||Li pouch cells in **a**, LiPF₆-EC/DEC, **b**, LiFSI-TEP/TTE, **c**, LiFSI-TEP/C3-2Cl.

In addition, to examine the decomposition products after long-term cycling for C-Cl bond stability, we have supplemented the TEM, XPS and TOF-SIMS characterizations for the CEI layer formed on the cycled 4.5 V NCM523 cathode. The results have been discussed in our revised manuscript.

To compare the difference in CEI layer at higher cut-off voltage, 4.5 V NCM523 cathodes were also disassembled and characterized from coin cells after 50 cycles in TEP-based electrolytes with/without C3-2Cl diluent (Supplementary Fig. 32). Compare to the CEI layer formed by LiFSI-TEP, a thin (~5 nm) and uniform CEI layer was also observed on the surface of 4.5 V NCM523 cathode in LiFSI-TEP/C3-2Cl, which consists of negligible Cl-contained species and less C/P-contained species derived by C3-2Cl diluent and TEP solvent. These results indicate that C3-2Cl diluent is also inert to be decomposed at high voltage, while it also benefits to inhibit the decomposition of TEP solvent.

Supplementary Fig. 32 | Characterizations of CEI layers formed on the 4.5 V NCM523 cathodes after 50 cycles. TEM of CEI layers formed by **a**, LiFSI-TEP/C3-2Cl and **b**, LiFSI-TEP. XPS spectra of CEI layers formed by **c**, **e**, **g**, LiFSI-TEP/C3-2Cl and **d**, **f**, **h**, LiFSI-TEP. TOF-SIMS of **i**, F^- , **j**, C_2HO^- and **k**, PO^- species in CEI layers formed by different electrolytes.

Point-by-point responses to reviewers' comments

Reviewer #1: *Accept.*

Response: We are very grateful to the reviewer for the recognition of our revised manuscript.

Reviewer #2: *The responses by Wang & Tu et al. to my critical comments/suggestions are mostly satisfactory. However, there are still some points remaining for the authors should consider for revision.*

Response: We deeply appreciate the reviewer for your recognition and valuable comments. Your kind reminder and professional suggestions are very helpful for us to improve the quality of our manuscript. Our point-by-point replies to the comments are detailed as follows.

Comment 1) *It is interesting that the proposed electrolyte is able to form a good performing SEI for lithium metal, but not able to form a SEI providing protection against graphite exfoliation. This is an important observation for the readers to be aware of, and the authors should clearly mention these observations in the manuscript (they can share the results in SI or not, as they prefer).*

Response: We deeply thank the reviewer for the precious suggestion. We have supplemented these results in our revised manuscript and SI.

In addition to analyzing the stability of the flame retardant electrolyte on the Li metal electrode, we also tried to apply it to the commercial Li-ion battery system with graphite (Gr) as the anode. First, we have assembled a NCM523||Gr pouch cell with Gr anode, using the LiFSI-TEP/C3-2Cl electrolyte with a charge-discharge rate of 0.05C at room temperature. Regrettably, this pouch cell showed a low discharge capacity at initial cycle (Supplementary Fig. 21a), indicating some undesired reactions are occurred in this system. Then, the cycled pouch cell was disassembled, showing obvious black reaction mottling and powder dropping phenomena on the surface of Gr electrode (Supplementary Fig. 21b). The X-ray diffraction (XRD) spectra demonstrate that the

intensity of typical Gr diffraction peak (002) at 26.5° significantly decreases after first cycle (Supplementary Fig. 21c), suggesting the structure of Gr is partially destroyed by the co-intercalation or reaction of the electrolyte. To further verify above conjecture, morphologies of the pristine and cycled Gr electrodes were observed by SEM. It is found that a compact and flat Gr particle was observed in pristine electrode (Supplementary Fig. 21d), but it turns to an exfoliated layered structure after cycling (Supplementary Fig. 21e). The above results prove that this kind of electrolyte is not suitable for Li-ion battery system with Gr anode at this stage, and further research on its reaction mechanism in this system is needed in the follow-up work.

Supplementary Fig. 21 | Study on LiFSI-TEP/C3-2Cl electrolyte in NCM523||Gr pouch cell. **a**, Initial charge-discharge profile of pouch cell at 0.05C. **b**, Optical image of cycled Gr anodes and separators assembled from pouch cell. **c**, XRD spectra of pristine and cycled Gr electrodes. SEM images of the morphology of **d**, pristine Gr and **e**, cycled Gr.

Comment 2) CV/LSV/CTTA tests. Fig3a. In this test, the authors also decided to scan below 0 V vs. Li^+/Li . As expected Li plating started in this region, resulting in high current, thereby masking in the graph the most relevant section above 0 V where side reactions occur at much lower currents. The most interesting region in such a CV/LSV experiment is the potential region above 0 V, where information about the stability of electrolyte could be achieved. The authors don't need to repeat the experiments, but

they should modify this graph with a new axis range for both x and y axis. They can give the data only for the first cycle where X-axis should only show 0 - 1 V region, and the y-axis should be scaled accordingly. They can still use the existing graph as an additional graph in MS or SI. Figure S10-a. Please double check the units of charge as it seems to be wrong.

Response: We sincerely thank the reviewer for the professional suggestions. We have supplemented the enlarged CV curves and relevant discussions in our revised manuscript and SI. Besides, since we previously calculated the accumulated charge per unit area based on the effective area of the electrode, the unit have been modified. In order to solve this problem, we have referred to the relevant literature (Nat. Commun. 2023, 14: 6946) and modified this mistake (as shown in Supplementary Fig. 10a). Detailed information is shown in below.

Cyclic voltammetry (CV) of Cu||Li cells was also measured to evaluate the electrochemical reduction behavior of electrolytes with different diluents. From Fig. 3g, it is observed that Li metal exhibits firstly increasing plating/stripping reversibility in electrolytes containing short-chain diluents from C1-2Cl, C2-2Cl, to C3-2Cl. However, when the carbon chain length of the diluents increases to C4-2Cl and C5-2Cl, Li metal reversible plating/stripping behavior gets worse, even much worse than LiFSI-TEP electrolyte. This phenomenon is consistent with above results, demonstrating that the electrolytes with C1-2Cl, C2-2Cl, C3-2Cl diluents show good electrochemical reduction stability, but the electrolytes with C4-2Cl or C5-2Cl diluents exhibit bad electrochemical reduction stability. The enlarged CV curves from 2.0 V to 0 V in the reduction process display that typical reduction peaks at ~1.2 V ascribed from LiFSI decomposition (Supplementary Fig. 13) ^[39]. No other significant reduction peak observed in the electrolytes with C4-2Cl or C5-2Cl diluents may be due to the electrochemical reduction peak overlaps with Li deposition process.

Supplementary Fig. 13 | Enlarged CV curves of Cu||Li cells with different electrolytes from 2.0 V to 0 V.

Supplementary Fig. 10 | a, Accumulated charges of Cu||Li cells with different electrolytes. Voltage-time curves of Cu||Li cells with b, LiFSI-TEP, c, LiFSI-TEP/C1-2Cl, d, LiFSI-TEP/C2-2Cl, e, LiFSI-TEP/C3-2Cl, f, LiFSI-TEP/C4-2Cl, g, LiFSI-TEP/C5-2Cl.

Reviewer #3: *The authors have improved the paper in many aspects, but still, a few major issues remain. Numbers in the following refer to those of my previous review.*

Response: We sincerely appreciate the reviewer for your recognition and valuable comments. Your kind reminder and professional suggestions are very helpful for us to

improve the quality of our manuscript. Our point-by-point replies to the comments are detailed as follows.

Comment 3) *Simply providing a snapshot of a commercial software tool for contact angle determination does not prove the claimed accuracy of the contact angle measurements. The authors should rather put forward strong arguments why they still report four significant figures or round the results to the nearest integer.*

Response: We would like to thank the reviewer again for your patient and precious reminder, and we are deeply sorry that we previously overtrusted the result of automatic fitting by the commercial software and ignored its accuracy. In order to be more rigorous, we have rounded the results to nearest integer in our revised manuscript and SI.

Supplementary Fig. 24 | Contact angle tests of different electrolytes. **a**, LiPF₆-EC/DEC, **b**, LiFSI-TEP, **c**, LiFSI-TEP/C3-2Cl, **d**, LiFSI-TEP/TTE.

Comment 8) *The name of the barostat is mentioned, but its control parameters are still missing.*

Response: We deeply thank the reviewer for your kind reminder about this negligence. We have carefully addressed this oversight in the revised manuscript by explicitly detailing the barostat settings.

The MD simulations were conducted using the COMPASSIII force field, with atom types and partial charges optimized based on prior validated studies.^[52–55] The system was subsequently equilibrated in the isothermal-isobaric (NPT) ensemble for 200 ps (picoseconds) with a time step of 1.0 fs (femtoseconds). To maintain thermodynamic stability, a Nosé-Hoover thermostat and a Parrinello-Rahman barostat were employed,^[56,57] with the following parameters: Cell time constant: 1.0 ps; Compressibility: $4.5 \times 10^{-5} \text{ GPa}^{-1}$; Coupling mode: Isotropic. The temperature was held constant at 298.15 K, and the pressure was maintained at 1 bar throughout the simulation.

Comment 9) *The authors now claim to have used COMPASSIII, whereas they reported to have used COMPASS II in the previous version of the paper. Please clarify. Also, please provide a reference where the actual values for the model parameters, e.g. those of Li^+ , have been reported. The provided screenshot is not helpful.*

Response: We sincerely appreciate the reviewer for your precious comments. To our best knowledge, Both COMPASS II and COMPASS III are all-atom force fields implemented in the Materials Studio Forcite module, sharing similar functional forms (e.g., bond stretching, angle bending, torsion, van der Waals, and electrostatics) for modeling organic molecules, polymers, and condensed-phase systems. However, COMPASS III represents a significant advancement over its predecessor, incorporating higher-level DFT-derived parameters to enhance the accuracy of short-range interactions (e.g., the Buckingham potential for metal-ligand and ion-solvent interactions) and refined charge assignments. In contrast, COMPASS II employs empirical Lennard-Jones potentials and fixed partial charges. This upgrade expands the force field's applicability to nanomaterials, metal-organic frameworks (MOFs), ionic liquids, and more complex systems like 2D materials and hybrid interfaces, offering enhanced accuracy for modern material simulations. Given the focus of this work on electrolyte systems (which often involve ionic species, polar solvents, and heterogeneous interfaces), we upgraded to COMPASS III to ensure robust predictions of thermodynamic and dynamic properties.

Additionally, we have supplemented more comprehensive references about COMPASSIII for the model parameters (particularly for Li⁺ interactions) in our revised manuscript. Detailed information is shown below:

52. Akkermans, R. L. C. et al. COMPASS III: automated fitting workflows and extension to ionic liquids. *Mol. Simula.* **47**, 540–551 (2021).
53. Efaw, C. M. et al. Localized high-concentration electrolytes get more localized through micelle-like structures. *Nat. Mater.* **22**, 1531–1539 (2023).
54. Wu, F. et al. Robust interphase derived from a dual-cation ionic liquid electrolyte enabling exceptional stability of high-nickel layered cathodes. *Energy Environ. Sci.* DOI: 10.1039/D5EE00669D (2025).
55. Kim, S. et al. Wide-temperature-range operation of lithium-metal batteries using partially and weakly solvating liquid electrolytes. *Energy Environ. Sci.* **16**, 11, 5108–5122 (2023).

Reviewer #4: *The authors have made efforts to address the previous concerns; however, significant contradictions and inconsistencies remain unresolved in both the revised manuscript and the provided rebuttal. These issues fundamentally undermine the validity of the conclusions and render the manuscript unsuitable for publication. Major concerns are outlined below:*

Response: We sincerely appreciate the reviewer for your thorough review and precious comments. However, we respectfully disagree with your partial comments, because these reviews ignore the differences brought about by different material systems and different test conditions. To address your concerns, we have supplemented sufficient experiments and given a more comprehensive discussion in our revised manuscript. Our point-by-point replies to the comments are detailed as follows.

Comment 1) *Contradiction Between Experimental Data and MD Simulation Results. The manuscript claims that chlorinated diluents participate in the inner solvation sheath of Li⁺ ions (~2 Å) based on MD simulations. However, this contradicts well-established literature (Zhang et al. *Nat. Commun.* 2022, 13, 5431) demonstrating that*

chlorinated diluents exhibit extremely weak coordination with Li⁺ (typical distances >4 Å). The authors' own experimental data further corroborate this inconsistency: the solubility of Li salts in chlorinated diluents is less than 6 mM (2 – 3 orders of magnitude lower than in the primary phosphate-based solvents). This suggests that chlorinated diluents are thermodynamically unfavorable for inner-shell coordination. In addition, the polar TEP molecule (triethyl phosphate), which has stronger coordination capability than ethers, would even further limit the coordination of chlorinated diluents with Li⁺.

Response: We would like to thank the reviewer for your precious comment, but we respectfully disagree it, because there are many existing literatures and our supplementary experiments can strongly support our previous results, the detailed reasons are shown below.

First, there have been a lot of literatures (*Energy Environ. Sci.* 2024, 17, 6079; *Energy Environ. Sci.* 2023, 16, 546) proving that DME molecule has a very high coordination ability with Li⁺ due to its special molecular structure, which causes the dioxygen atoms in DME molecule to produce chelating effect with Li⁺. By comparing the binding energy of TEP molecule to Li⁺ reported in the literatures (*ACS Energy Lett.* 2024, 9, 1, 136–144; *Adv. Funct. Mater.* 2023, 33, 2212605), it was found that DME molecule shows higher binding energy than TEP molecule due to its special chelation with Li⁺. Meanwhile, our DFT calculation results also prove that TEP molecule shows lower binding energy (-235.0 kJ mol⁻¹) with Li⁺ than that of DME molecule (-244.9 kJ mol⁻¹), as shown in **Figure R1a**.

Second, due to the different binding ability and binding structure of TEP and DME to Li⁺, the competitive coordination between diluent and Li⁺ will be different. The structure of the chelated coordination of DME molecules with Li⁺ not only has higher binding force but also has larger steric hindrance, thus theoretically making it more difficult for the diluent molecules to participate in the Li⁺ coordination. In order to prove the above phenomenon, the RDF curves between Li⁺ and the Cl of the C2-2Cl diluent in the LiFSI-TEP/C2-2Cl and LiFSI-DME/C2-2Cl electrolytes were obtained by MD simulations to determine the coordination of the diluent molecules and Li⁺ in

different solvent systems. As shown in **Figure R1b**, it can be obviously found that the typical coordination distance of diluent molecules and Li^+ in the DME-based electrolyte is around 6 Å, which is consistent with the results reported in the literature (*Nat. Commun.* 2022, 13, 5431). However, the diluent molecules in the TEP-based electrolyte are more likely to appear in the ~ 2.5 Å surrounding Li^+ , proving that more diluent molecules participate in the inner solvation structure of Li^+ .

Figure R1 | **a**, Binding energies of Li^+ -TEP and Li^+ -DME obtained by DFT calculations. **b**, RDF curves of Li^+ coordinated with Cl in C2-2Cl diluent obtained by MD simulations in LiFSI-TEP/C2-2Cl and LiFSI-DME/C2-2Cl electrolytes.

Third, to our best knowledge, the dissolution of Li salt requires the solvent to overcome the energy barrier of the strong ionic bond between Li^+ and anion, this process is often difficult and therefore requires a strongly polar solvent to ensure a high solubility of Li salt. This indicates that the low solubility of weakly polar solvent to Li salt does not mean that it not participates in the Li^+ solvation structure, because when the energy barrier between Li^+ and anion is overcome by using a strongly polar solvent, the competitive coordination between the weakly polar solvent and the strongly polar solvent will be easier, so it may lead to its participation in the Li^+ solvation structure. And there's a lot of literature out there that supports this (*Adv. Funct. Mater.* 2021, 31, 2005991; *Angew. Chem. Int. Ed.* 2024, 63, e202317176; *Nat. Commun.* 2024, 15, 6292), e.g. fluorobenzene (FB) or 1,2-difluorobenzene (DFB) diluents have been reported to be almost insoluble to Li salts, with little involvement in the Li^+ solvation structure in strongly coordinated solvents (such as DME), but partially involved in the inner solvation structure of Li^+ in phosphate solvents (such as TMP).

Comment 2) *Negligible Impact of Diluent Coordination on Electrochemical Performance. Even if a minor fraction of chlorinated diluents coordinates with Li^+ (which remains unproven), the proportion would be vanishingly small due to their ultra-low solubility and weak coordination strength. Such a negligible contribution cannot plausibly account for the observed electrochemical improvements (e.g., cycling stability or Li-metal compatibility). Instead, the intrinsic reactivity differences between uncoordinated diluent molecules (e.g., stability against Li metal or electrolyte decomposition pathways) are far more likely to dominate performance outcomes. Regrettably, the authors entirely omit discussions or experiments addressing these critical factors, rendering their mechanistic interpretation incomplete and potentially misleading.*

Response: We sincerely thank the reviewer for your precious comment, and we are very sorry for your misunderstanding caused by our omission of some key experiments. To address your concerns, we have supplemented sufficient experiments and discussions in our revised manuscript, the results also strongly support our previous conclusions. The detailed reasons are shown in below.

First, as reported in a large number of relevant literatures (*ACS Energy Lett.* 2022, 7, 490–513; *Adv. Mater.* 2023, 35, 2206009; *Energy Mater* 2021, 1, 100004; *Interdiscip. Mater.* 2023, 2, 833–854), the Li^+ solvation structure in the electrolyte can significantly affect the electrode interfacial behavior, which plays a crucial role in battery performance. Particularly, the solvent molecules involved in the Li^+ solvation structure are more likely to migrate to the electrode surface together with Li^+ and have electron gain and loss behavior, thus forming SEI film. This process tends to consume Li^+ and solvent molecules until the formation of a dense and stable SEI film to isolate electron transport. However, if there are unstable components in the Li^+ solvation structure, which cannot participate in the formation of stable SEI film components after reaching the electrode interface, the reaction will continue and the Li source or electrolyte will be consumed, thus leading to large polarizations and battery failure. Therefore, the Li^+ solvation structure is crucial, and the presence of even a small amount of unstable

components can lead to large differences in electrode interface and battery performance, which is consistent with our conclusions.

Second, in order to understand more clearly whether the reduction of coordinated diluent or the free diluent plays a dominant role in the battery performance, we have supplemented the relevant experiments and discussions in our revised manuscript. The results also show that the uncoordinated diluents are stable with Li metal, but the long-chain diluent coordinated Li^+ complex is easily reduced to cause the deterioration of electrode interface, which is consistent with the dead Li, LiCl and organic species detected on Li metal surface by SEM, XPS and TOF-SIMS results in our previous manuscript. Detailed information is shown in below.

All above results demonstrate the diluents with different carbon-chain length significantly affect the stability of electrolytes and Li metal. In order to figure out the reason, we firstly investigated the chemical reactions between the different diluents and Li metal. Fresh Li metal sheets were separately soaked in different pure diluents for 2 weeks. Supplementary Fig. 11 obviously displays that after soaking Li metal in five pure diluents for 2 weeks, all the diluents still remain transparent and the surfaces of five Li metal sheets still present bright metallic luster. The soaking experiments indicate that these diluents from C1-2Cl to C5-2Cl are chemically stable with Li metal.

Cyclic voltammetry (CV) of Cu||Li cells was also measured to evaluate the electrochemical reduction behavior of electrolytes with different diluents. From Fig. 3g, it is observed that Li metal exhibits firstly increasing plating/stripping reversibility in electrolytes containing short-chain diluents from C1-2Cl, C2-2Cl, to C3-2Cl. However, when the carbon chain length of the diluents increases to C4-2Cl and C5-2Cl, Li metal reversible plating/stripping behavior gets worse, even much worse than LiFSI-TEP electrolyte. This phenomenon is consistent with above results, demonstrating that the electrolytes with C1-2Cl, C2-2Cl, C3-2Cl diluents show good electrochemically reduction stability, but the electrolytes with C4-2Cl or C5-2Cl diluents exhibit bad electrochemical reduction stability. The enlarged CV curves from 2.0 V to 0 V in the reduction process display that typical reduction peaks at ~ 1.2 V ascribed from LiFSI decomposition (Supplementary Fig. 13) ^[39]. No other significant reduction peak

observed in the electrolytes with C4-2Cl or C5-2Cl diluents may be due to the electrochemical reduction peak overlaps with Li deposition process.

Supplementary Fig. 11 | Optical images of the state of Li metal sheets soaked in different diluents after 2 weeks.

Supplementary Fig. 13 | Enlarged CV curves of Cu||Li cells with different electrolytes from 2.0 V to 0 V.

Comment 3) *Discrepancies in ARC Thermal Runaway Results.* The manuscript reports that cells using chlorinated diluents or TTE exhibit suppressed thermal runaway (no significant dT/dt increase), which starkly conflicts with recent studies on phosphate ester-based localized high-concentration electrolytes (LHCEs). Multiple independent

works (including systems with TEP solvents or TTE diluents) consistently demonstrate that thermal runaway persists in such systems. The authors' anomalous results raise serious concerns about the validity of their testing methodology. These unresolved discrepancies cast doubt on the reliability of the ARC data and the broader conclusions drawn from them.

Ref 1. Chen et al. Advanced Materials. 2024, 36, 2312302.

Ref 2. Ouyang et al. Nat. Commun. 2020, 11, 5100.

Ref 3. Li et al. Journal of Energy Chemistry, 2024, 97, 156.

Response: We deeply thank the reviewer for your precious comment, and we are very sorry for your confusion caused by our incomplete analysis before. Actually, the differences of these results were caused by the differences in battery design parameters and the contents of electrolyte, which is proved by our supplemented experiments in our revised manuscript. Detailed discussions are shown in below.

Notably, the above results are inconsistent with those reported in previous papers that the use of flame retardant electrolytes based on phosphate solvents and TTE diluent will also cause thermal runaway^[43,44], which may be directly related to the differences in battery design parameters and the contents of electrolyte. To fig out the reasons, we refer to the previous paper and increase the electrolyte injection amount from 1.5 $\mu\text{L mAh}^{-1}$ to 3 $\mu\text{L mAh}^{-1}$ (close to 3 g Ah^{-1} used in previous papers)^[44,45]. As shown in Supplementary Fig. 39d and e, when the amount of electrolyte is high, the thermal runaway of the battery still occurs even if the flame retardant electrolyte is used, which is consistent with the previous reports. Compare to LiFSI-TEP/TTE electrolyte, the pouch cell with LiFSI-TEP/C3-2Cl electrolyte shows higher T1 temperature and lower gas production. Meanwhile, as shown in Fig. 6g, a T2 temperature of 202.1 $^{\circ}\text{C}$ was detected in the pouch cell with 3 $\mu\text{L mAh}^{-1}$ LiFSI-TEP/C3-2Cl, which is much higher than the T2 values obtained in 3 $\mu\text{L mAh}^{-1}$ LiFSI-TEP/TTE (154.3 $^{\circ}\text{C}$) and even 1.5 $\mu\text{L mAh}^{-1}$ EC/DEC-based electrolyte (104.3 $^{\circ}\text{C}$). Above results strongly suggest that the flame-retardant electrolyte with C3-2Cl diluent can significantly enhance the safety of LMBs. In addition, even under extreme safety test conditions of nail penetration short circuit (Supplementary Fig. 42) or overcharge abuse (Supplementary Fig. 43), the

pouch cell with $1.5 \mu\text{L mAh}^{-1}$ LiFSI-TEP/C3-2Cl can still guarantee low temperature rise with no apparent thermal runaway, demonstrating high safety for practical applications. According to the above results, a new finding is also reported in this work, i.e., the synergistic effect of flame retardant electrolyte and low electrolyte injection amount helps to suppress the thermal runaway problem of LMBs, which provides an effective reference for the design of high safety and practical LMBs.

Supplementary Fig. 39 | ARC results with temperature-time and gas production-time curves of NCM83||Li pouch cells in **a**, LiPF₆-EC/DEC with an electrolyte amount of $1.5 \mu\text{L mAh}^{-1}$, **b**, LiFSI-TEP/TTE with an electrolyte amount of $1.5 \mu\text{L mAh}^{-1}$, **c**, LiFSI-TEP/C3-2Cl with an electrolyte amount of $1.5 \mu\text{L mAh}^{-1}$. **d**, LiFSI-TEP/TTE with an electrolyte amount of $3 \mu\text{L mAh}^{-1}$, **e**, LiFSI-TEP/C3-2Cl with an electrolyte amount of $3 \mu\text{L mAh}^{-1}$.

Fig. 6 | **g**, ARC tests of NCM83||Li pouch cells with different electrolytes and different electrolyte amounts.

Point-by-point responses to reviewers' comments

Reviewer #2: *I appreciate the revisions made by the authors. The responses are partially successful in resolving some concerns I have had, however, new results the authors provided also came with some new confusion/concerns. “I vs. E” data from CV tests for 0-2V region, which is now visible in the new graph, shows that LiFSI-TEP, C4-2Cl and C5-2Cl electrolytes have the highest stability against reduction. On the other hand, other electrolytes (including C3-2Cl) seem to be more reactive, with reactivity increasing from C3 to C2 and C1. This is not in agreement with the trends of other electrochemical tests. Both from the CV and CTTA data, it seems like that the LiFSI-TEP electrolyte is the most stable electrolyte from reduction stability perspective. The authors state in their conclusion that “C3-2Cl diluent shows outstanding redox stability …”. Even though it is clear that this electrolyte results in good cycling performance in Li-Li cells and NCM-Li cells, it is not clear if a better electrochemical stability of this electrolyte is responsible for the better performance of symmetrical cells and half cells with this electrolyte. The authors also stated that the proposed electrolyte is not able to passivate a graphite electrode. Therefore, I would recommend the authors to provide more support with more comprehensive electrochemical analysis on their claims about the reduction stability of their electrolytes. This would require: i) more comprehensive CE tests in full cells with limited Li inventory (e.g., NCM-Cu anode free cells, NCM-Si full cells, etc.) performed at RT and elevated temperature at low C-rates, ii) CTTA tests performed directly (i.e., without high current/charge initial step as authors applied), iii) potentiostatic holding tests in anode-free cells to observe the charge accumulation at a potential near the lithium metal potential, e.g., holding at 0.1 V for 50 hours, etc.*

Response: We deeply appreciate the reviewer for your valuable comments. Your precious suggestions are very helpful for us to improve the quality of our manuscript. We have supplemented the experiments according to the comments and the detailed replies are shown in the revised manuscript.

Firstly, NCM-Cu anode free cells with different electrolytes were assembled and the results show as follows: more comprehensive CE tests in NCM523 (areal loading of 2.0 mAh cm^{-2})||Cu full cells with limited Li inventory were performed (Supplementary Fig. 9). As expected, at initial 0.05C charge-discharge process, the NCM523||Cu full cell with LiFSI-TEP/C3-2Cl exhibits the highest ICE (83.3%) and the highest initial discharge capacity (170.9 mAh g^{-1}). Meanwhile, it also displays the most stable cycling performance with the highest capacities and the highest CEs ($> 97.4\%$) at 0.1C, indicating that the C3-2Cl diluent has a best electrochemical stability compare to other C-2Cl diluents.

Supplementary Fig. 9 | **a**, Initial charge-discharge profiles and Initial CEs of NCM523||Cu cells with different electrolytes at 2.8~4.3 V and 0.05C. **b**, Discharge capacity and **c**, Coulombic efficiency of NCM523||Cu cells with different electrolytes at 2.8~4.3 V and 0.1C.

Secondly, CTTA tests were retested directly without high current/charge initial step. The experimental details and the new results are shown in the revised manuscript and supporting information. As shown below:

CTTA was performed using Cu||Li cells. During the titration step, a current of $I(\tau)$

= 10 $\mu\text{A cm}^{-2}$ was applied for 0.1 h, corresponding to a charge of $q(\tau) = 1 \mu\text{Ah cm}^{-2}$ was deposited on Cu foil. After each titration step, an open circuit voltage (OCV) period began, during which the cell voltage (E) was monitored. As the titrated Li was gradually consumed by side reactions, the E gradually increased and the OCV period was terminated when $E = 0.05 \text{ V}$. The whole process with titration step and the OCV period were repeat until 400 h.

Coulometric titration time analysis (CTTA) with Cu||Li cell was performed to quantify the side reactions between electrolyte and Li metal electrode [40]. The potential profiles of the various electrolytes during the 400 h test and the total accumulated charge consumed in side reactions (q_{Σ}) were shown in Supplementary Fig. 11. The order of accumulated charge consumed after 400 h was: LiFSI-TEP/C4-2Cl > LiFSI-TEP/C5-2Cl > LiFSI-TEP/C1-2Cl > LiFSI-TEP/C2-2Cl > LiFSI-TEP > LiFSI-TEP/C3-2Cl. This indicates that C4-2Cl and C5-2Cl exhibit the highest reactivity with Li metal anode, while C3-2Cl demonstrates the highest stability among all symmetric dichloroalkane diluents.

Supplementary Fig. 11 | a, Accumulated charges of Cu||Li cells with different electrolytes. Voltage-time curves of Cu||Li cells with b, LiFSI-TEP, c, LiFSI-TEP/C1-2Cl, d, LiFSI-TEP/C2-2Cl, e, LiFSI-TEP/C3-2Cl, f, LiFSI-TEP/C4-2Cl, g, LiFSI-TEP/C5-2Cl.

Thirdly, potentiostatic holding tests were also carried out according to your suggestion. The LiFSI-TEP/C3-2Cl also displays the lowest accumulated charge, showing the highest electrochemical stability. As shown below:

To verify the above conclusion, the Cu||Li cells were also assembled for

potentiostatic holding tests under the conditions of 0.1 V (eliminate the interference caused by Li reduction) for 50 h. As shown in Supplementary Fig. 12, after 50 h of potentiostatic holding, the accumulated charge of the Cu||Li cell with LiFSI-TEP/C3-2Cl is 16.9 μAh , which is lower than those of TEP-based electrolytes with other C-2Cl diluents and the LiFSI-TEP electrolyte, verifying the highest electrochemical stability of the C3-2Cl diluent.

Supplementary Fig. 12 | Accumulated charges of Cu||Li cells with different electrolytes at a 0.1 V potentiostatic holding for 50 h.

In addition, we respectfully **don't agree with the reviewer's opinion** that “I vs. E data from CV tests for 0-2V region shows that LiFSI-TEP, C4-2Cl and C5-2Cl electrolytes have the highest stability against reduction.”

Firstly, according to numerous literature reports (*J. Chem. Educ.* **2018**, 95, 197; *Electrochem.* **2022**, 90, 10, 102005; *Anal. Chem.* **2005**, 77, 20, 6702), the response current of the CV test is affected by various factors, such as the viscosity, conductivity, and diffusion coefficient of the electrolyte. Due to the high viscosity and low ionic conductivity of the LiFSI-TEP electrolyte, it results in a relatively low response current density in CV curve. Therefore, the CV results show that within such a low current response range (less than 6 $\mu\text{A cm}^{-2}$), the differences in the reduction peaks will be

severely interfered by the physicochemical properties of the electrolyte and the precision of the instrument, and are not necessarily solely determined by the reduction behavior of the electrolyte.

Secondly, the enlarged CV curves from 2.0 V to 0 V in the reduction process display that typical reduction peaks at ~ 1.2 V ascribed from LiFSI decomposition (Supplementary Fig. 15)^[39], which is consistent with the reports in the literature (*Angew. Chem. Int. Ed.* **2022**, e202115909; *J. Phys. Chem. C* **2018**, 122, 18, 9825; *Energy Storage Mater.* **2019**, 23, 350). The reduction of FSI⁻ anions is related to the special solvation structure of high-concentration/localized high-concentration electrolytes, proving that such electrolytes are conducive to the formation of an anion-derived SEI film on the electrode surface, which has been proven to be helpful in suppressing interface side reactions and the growth of lithium dendrites. This conclusion is also consistent with the results of XPS and TOF-SIMS in our work, demonstrating higher contents of fluorides on the Li electrode surface.

Thirdly, the whole CV curves in Fig. 3g clearly prove Li metal reversible plating/stripping behavior is much better in the electrolyte with C3-2Cl diluent than those in the electrolytes with C4-2Cl and C5-2Cl diluents, proving the highest stability in C3-2Cl-contained electrolyte.

In summary, all the supplemental experiment results also prove our conclusion that the C3-2Cl diluent shows outstanding redox stability.